

# Comparison of hybrid schemes for the combination of Shallow Approximations in numerical simulations of the Antarctic Ice Sheet

Jorge Bernales[1,2], Irina Rogozhina[3,1], Ralf Greve[4], and Maik Thomas[1,2]

[1]GFZ German Research Centre for Geosciences, Section 1.3: Earth System Modelling, Potsdam, Germany
[2]Institute of Meteorology, Free University Berlin, Berlin, Germany
[3]MARUM Centre for Marine Environmental Sciences, University of Bremen, Bremen, Germany
[4]Institute of Low Temperature Science, Hokkaido University, Sapporo, Japan

*Correspondence to:* Jorge Bernales (bernales@gfz-potsdam.de)

**Abstract.** The Shallow Ice Approximation (SIA) is commonly introduced in ice-sheet models to simplify the equations describing ice dynamics. However, the SIA is not applicable in fast flowing regions where basal sliding operates. To overcome this limitation, recent studies have introduced heuristic, hybrid combinations of the SIA and the Shelfy Stream Approximation. Here, we implement four different hybrid schemes into a model of the Antarctic Ice Sheet in order to compare their performance under a realistic scenario. For each scheme, the model is calibrated using an iterative technique to infer the spatial variability in basal sliding parameters. Model results are validated against topographic and velocity data. Our analysis shows that the calibration compensates for the differences between the schemes, producing similar ice sheet configurations through quantitatively different parameter distributions. Despite this, we observe a robust agreement in the reconstructed patterns of hard vs. soft basal conditions. We use averaged and swapped parameter distributions to demonstrate that the results of the model calibration cannot be straightforwardly transferred to models based on different approximations of ice dynamics. However, this requirement for internal consistency can be fulfilled through the implementation of easily adaptable calibration techniques, as shown in this study.

## 1 Introduction

Accurate projections of ice-sheet–driven sea level changes require the use of numerical models that are capable of capturing the dynamics of rapidly flowing regions and grounding-line zones (Pattyn et al., 2013). This requirement can be best accommodated using the most complete models which are currently available for modelling ice dynamics, referred to as Full Stokes models (FS) (e.g., Favier et al., 2012). However, time scales over which ice sheets build up and disintegrate in response to variations in the climatic forcing typically involve tens of thousands of years. Numerical experiments over such time spans are necessary, e.g., to separate the long-term transient component from relatively fast fluctuations in the ice volume during the observational record. These long-term, continental-scale paleo-simulations are currently unfeasible using FS models due to the computational expenses triggered by the non-linearity of the model equations and the complex interdependence of the involved quantities.





To overcome the contemporary spatio-temporal limitations of FS models, a hierarchy of approximations has been developed over the last decades. These approximations build upon neglecting terms in the momentum balance equations, and are categorised according to the degree of simplification (e.g., Hindmarsh, 2004). The simplest and most commonly used approximation for long-term simulations of ice sheet dynamics is the Shallow Ice Approximation (SIA) (e.g., Hutter, 1983). This approximation keeps only the gravity-driven vertical shear stress, predicting reasonably well the behaviour of ice masses which are characterised by a thickness much smaller than their horizontal dimensions, smooth bed topography, and minimal sliding at the base. However, this is not the case for areas of rapid ice flow, known as ice streams. Such areas serve as a bridge between grounded and marine portions of an ice sheet, where ice flow is often enhanced by basal conditions favourable for sliding. In these ice sheet sectors, membrane stresses become increasingly important, sharing many similarities with the floating extensions of ice sheets known as ice shelves. Ice floating in the sea water experiences almost no friction, and its behaviour is usually described by the Shallow Shelf Approximation (SSA) (Morland, 1987), which omits the vertical shear stress from the FS equations and neglects the basal drag. It is also important to note that the absence of a membrane stress transfer in the SIA renders this approximation invalid for modelling the grounding-line migration, that is the migration of an interface between grounded and floating ice sectors (Pattyn et al., 2012).

The limitations of SIA models in these highly dynamical regions have prompted the development of more sophisticated methods that have been designed to overcome these limitations, while still aiming at low computational costs. The approaches used by this new generation of continental-scale ice-sheet models include depth-integrated Blatter–Pattyn models based on the asymptotic analysis by Schoof and Hindmarsh (2010), algorithms that detect and apply the SIA where it is valid and use the FS elsewhere (Ahlkrona et al., 2015), and so-called hybrid models utilising heuristic combinations of the SIA and the Shelfy Stream Approximation (SStA, which is the SSA including basal drag) (Bueler and Brown, 2009; Winkelmann et al., 2011; Pollard and DeConto, 2012a). These hybrid models detect zones of potential fast flow where ice streams may operate and heuristically weight contributions of each approximation based on predefined criteria. The use of hybrid models enables simulations over hundreds of thousands of years on continental scales, yet showing a reasonable performance versus higher-order models in idealised scenarios and inter-comparison tests (e.g., Pattyn et al., 2013; Feldmann et al., 2014). Since the superposition of the SIA and the SStA is based on heuristics, the approaches used to combine the two approximations vary from model to model, ranging from weighted averages of both velocity solutions to a simple summation over the entire domain. Since in most cases the resulting velocity fields will be different, the possibility of having two different models yielding similar results under similar forcing could be attributed to other factors, such as the selection of additional model parameters, model calibration and initialisation, and a range of model-dependent biases.

Despite the above differences among models, all of them are subject to common limitations when applied to realistic scenarios. These result from the scarcity of observational data needed to reduce the errors introduced by poorly constrained model parameters. Mechanical properties of the subglacial bedrock surface and the associated distribution of zones where sliding can be enhanced by soft-bed conditions may serve as an example of such poorly constrained parameters. This is currently considered to be a major source of large, widespread misfits (several hundreds of metres) between the observed and modelled elevations of the Antarctic Ice Sheet (AIS). Recent studies have attempted to quantify potential distributions of these



intrinsic bed properties using sophisticated inverse methods (e.g., Joughin et al., 2009; Morlighem et al., 2010; Arthern and Gudmundsson, 2010; Pralong and Gudmundsson, 2011; Arthern et al., 2015). These diagnostic methods focus mainly on the fit between the modelled and observed ice velocities. Pollard and DeConto (2012b) presented a much simpler algorithm, aiming to fit the observed surface elevations instead of velocities. The prognostic model is run forward in time, and the local elevation

error is used to periodically adjust the basal sliding parameters until the best fit between the observed and modelled elevations is attained. This procedure has the ability to drastically reduce large elevation errors during the calibration and initialisation of ice-sheet models, which is an important requirement for paleo-simulations that would otherwise be undermined by poor parameter choices.

In this paper, we evaluate the performance of four different hybrid approaches implemented as part of the same continental-

10 scale ice-sheet model, applied to the entire AIS. To this end, a calibration procedure based on the aforementioned iterative technique of Pollard and DeConto (2012b) is applied to each hybrid scheme using state-of-the-art observational data sets as constraints. For comparison purposes, the same procedure is carried out using a SIA-only model. The results of these experiments are validated against an independent, observational data set of surface ice velocities. Additionally, we shed light onto the ways in which relative contributions of the shallow approximations are derived by different hybrid schemes. Discrepancies

between the inferred distributions of basal sliding parameters are used to quantify the effects of the high uncertainty in basal mechanical properties. For hybrid approaches involving adjustable parameters, we also explore the sensitivity of the results to parameter variations.

This work is structured as follows. First, the ice-sheet model and the hybrid schemes are described in Section 2, where we also detail the adapted inverse technique for the calibration of the basal sliding parameters. The setup of the experiments can be

found in Section 3. The results are presented and discussed in Section 4, followed by the summary and conclusions provided in Section 5.

## 2 Methods

### 2.1 Model overview

In this study, the simulations of the AIS are carried out using the open source, three-dimensional, thermo-mechanical ice

sheet-shelf model SICOPOLIS (SImulation COde for POLythermal Ice Sheets) version 3.2-dev, revision 619 (Greve, 1997; Greve and Blatter, 2009; Sato and Greve, 2012). It uses finite differences to solve the numerically approximated SIA and SSA equations for grounded and floating ice, respectively. Relevant modifications introduced in the model specifically for this study are presented in a greater detail in Sections 2.2 and 2.3.

The model considers ice sheets as polythermal, i.e., explicitly keeps track of potential temperate regions in which ice is at the

30 pressure-melting point (Greve, 1997). Within these regions, ice and small amounts of liquid water can coexist, and the water content is used as an additional input for a regularized Glen's flow law (Glen, 1955) utilised in our experiments, following



Greve and Blatter (2009), in the form

$$\eta(T_{\mathrm{m}}, \sigma_{\mathrm{e}}) = \frac{1}{2EA(\sigma_{\mathrm{e}}^{n-1} + \sigma_0^{n-1})}, \tag{1}$$

where $\eta$ denotes ice shear viscosity, $T_{\mathrm{m}}$ is the temperature below the pressure-melting point, $\sigma_{\mathrm{e}}$ is the effective shear stress, $\sigma_0 = 10\,\mathrm{kPa}$ is a small constant used to prevent singularities when $\sigma_{\mathrm{e}}$ is very small, $n = 3$ is the flow law exponent, and $A$ is

a temperature- and pressure-dependent rate factor (Cuffey and Patterson, 2010). In the temperate ice regions, $A$ is modified to account for the aforementioned water content, following Lliboutry and Duval (1985). The empirical coefficient $E$ is a flow enhancement factor, which is introduced to account for the effects of anisotropic ice fabric. The value of $E$ depends on the deformation regime of ice, thus depending on whether it is being applied on grounded ice, where horizontal shear prevails, or floating ice, dominated by longitudinal stretching (Ma et al., 2010). In general, large-scale marine ice-sheet models use a

homogeneous, constant value ranging between $1$ and $8$ for the grounded ice, and between $0.2$ and $1$ for ice shelves (e.g., de Boer et al., 2015). Within these ranges, the computed age of the ice is often used to assign different values for glacial and interglacial ice. Here, we use $E = 1$ and $E = 0.5$ for grounded and floating ice, respectively. These values are smaller than those chosen in previous studies using SICOPOLIS (e.g., Sato and Greve, 2012), and are based on our initial tests and the sensitivity analysis by Pollard and DeConto (2012b).

At the base of the grounded ice sectors, bedrock stress conditions and the associated potential for sliding are linked to the basal SIA velocity, $\boldsymbol{u}_b$, through an empirical Weertman-type sliding law (Weertman, 1964; Dunse et al., 2011), in the form

$$\boldsymbol{u}_{\mathrm{b}} = -\frac{C_{\mathrm{b}}}{N_{\mathrm{b}}^q} |\boldsymbol{\tau}_{\mathrm{b}}|^{p-1} \boldsymbol{\tau}_{\mathrm{b}}, \tag{2}$$

where $\boldsymbol{\tau}_{\mathrm{b}}$ is the basal shear stress, $N_{\mathrm{b}}$ is the effective basal pressure, and $p = 3$ and $q = 2$ are the sliding law exponents. The parameter $C_{\mathrm{b}}$ depends on the basal temperature and pressure conditions:

$$C_{\mathrm{b}} = C_0 e^{T_{\mathrm{m}}/\gamma}, \tag{3}$$

where the exponential function controls the amount of sliding, depending on the temperature below the pressure-melting point $T_{\mathrm{m}}$ and the sub-melt-sliding parameter $\gamma$, ensuring a smooth transition across different basal thermal conditions. A spatially varying factor $C_0$ is introduced to account for differences in the bedrock material properties affecting sliding (e.g. hard bedrock vs. water-saturated sediments). Potential distributions of $C_0$ have been explored using different iterative and inverse approaches

and a variety of sliding laws, aiming to find spatially varying values that minimise the discrepancies between the modelled and observed quantities such as ice thickness, ice surface velocity and elevation change (e.g., Joughin et al., 2009; Morlighem et al., 2010; Arthern and Gudmundsson, 2010; Pralong and Gudmundsson, 2011; Pollard and DeConto, 2012b; Arthern et al., 2015)). A particular inverse technique implemented in our model is described in Section 2.3. Other model components include evolution equations for ice temperature and ice thickness, with the latter forced by independent modules for the computation

of the surface mass balance, basal melt, and changes in bedrock elevation (Greve and Blatter, 2009; Sato and Greve, 2012). A summary of the model parameters used in this study is provided in Table 1.





## 2.2 Hybrid schemes

For this study, four hybrid approaches have been implemented into the model. Each of them offers a different way to detect the fast flowing zones and combine the horizontal SIA velocity, $\boldsymbol{u}$, and the horizontal SStA velocity, $\boldsymbol{v}$. For each scheme, individual velocity solutions from each shallow approximation are calculated independently. It is important to note that SStA velocities in grounded ice regions include basal drag. This implies a major difference from SSA velocities computed in the floating ice shelf sectors, for which the friction at the ice-ocean interface is negligible.

For consistency, the basal drag term that enters the SStA equations is computed using the same sliding law as described in Section 2.1 (Eq. (2)), in the form

$$\boldsymbol{\tau}_{\mathrm{b}} = -\beta_{\mathrm{drag}}\boldsymbol{v}_{\mathrm{b}}, \tag{4}$$

where $\boldsymbol{v}_{\mathrm{b}}$ is the the basal SStA velocity and the drag coefficient, $\beta_{\mathrm{drag}}$, is computed as

$$\beta_{\mathrm{drag}} = \frac{N_{\mathrm{b}}^{\frac{q}{p}}}{C_{\mathrm{b}}^{\frac{1}{p}}} \left( \frac{1}{\sqrt{v_{\mathrm{b_x}}^2 + v_{\mathrm{b_y}}^2 + v_0^2}} \right)^{1-\frac{1}{p}}. \tag{5}$$

Here, $v_{\mathrm{b_x}}$ and $v_{\mathrm{b_y}}$ are the horizontal components of the SStA velocity at the ice base, and $v_0 = 0.01\,\mathrm{m\,a}^{-1}$ is a small regularisation quantity introduced to prevent singularities at the locations where there is no basal sliding (Bueler and Brown, 2009). Equation (5) allows for a consistent use of the inverted distributions of $C_0$ during the computation and combination of the SIA and SStA velocities in the grounded ice zones (Section 2.3).

The first hybrid scheme (henceforth HS-1) is the original implementation in SICOPOLIS v3.2-dev (revision 619) based on the slip ratio of grounded ice, computed as

$$r = \frac{|\boldsymbol{u}_{\mathrm{b}}|}{|\boldsymbol{u}_{\mathrm{s}}|}, \tag{6}$$

where $\boldsymbol{u}_{\mathrm{b}}$ (Eq. (2)) and $\boldsymbol{u}_{\mathrm{s}}$ are the basal and surface SIA velocities, respectively. At each iteration, and for each velocity component, the local slip ratio $r$ is compared to a prescribed threshold $r_{\mathrm{thr}}$ ranging from 0 to 1 (Section 3). If $r$ is larger than the threshold, that grid point is flagged as a streaming point and enters the computation of SStA velocities. Once SStA velocities are computed, the individual contributions from the SIA and SStA at each streaming grid point are determined using the weight

$$w(r) = \frac{r - r_{\mathrm{thr}}}{1 - r_{\mathrm{thr}}}, \tag{7}$$

which is assigned to the SStA. Then, for each streaming grid point, the hybrid horizontal velocity $\boldsymbol{U}$ is computed as

$$\boldsymbol{U} = w \cdot \boldsymbol{v} + (1 - w) \cdot \boldsymbol{u}, \tag{8}$$

recalling that $\boldsymbol{u}$ and $\boldsymbol{v}$ are the horizontal SIA and SStA velocities, respectively.



The second approach (henceforth HS-2) is based on the idea by Bueler and Brown (2009), in which SStA velocities are calculated over the entire ice sheet and used as a sliding velocity complementing a non-sliding SIA model. Velocities are computed as in Eq. (8), and the weighting function is adopted from Bueler and Brown (2009, Eq. (22))

$$w(|\boldsymbol{v}|) = \frac{2}{\pi}\arctan\left(\frac{|\boldsymbol{v}|^2}{v_{\mathrm{ref}}^2}\right), \tag{9}$$

where $v_{\mathrm{ref}}$ is a reference ice velocity (Section 3). The velocity $v_{\mathrm{ref}}$ marks the point for which the SIA and SStA contributions are equally weighted, i.e., the resulting hybrid velocity is a standard mean of both solutions. The weighting function is smooth, monotone, and its value converges towards 0 for small velocities and towards 1 when $\boldsymbol{v}$ is large compared to the reference velocity $v_{\mathrm{ref}}$. As in the HS-1, $w$ is used to compute respective contributions from the SIA and SStA, with the difference that Eq. (9) uses $v_{\mathrm{ref}}$ as the only criterion to determine the SStA contribution. The SStA velocities are calculated over the entire ice

sheet, and an a priori identification of fast flow zones is not required.

As described in Section 2.1, the SIA solution in SICOPOLIS already contains a sliding component coming from Eq. (2). To assess the influence of a sliding SIA, we have divided this hybrid scheme into two: A non-sliding-SIA sub-scheme (HS-2a) that replicates the idea of Bueler and Brown (2009), and a sliding-SIA sub-scheme (HS-2b) that keeps the sliding-SIA component and uses it to compute a slightly modified weight:

$$w(|\boldsymbol{u}_{\mathrm{b}}|) = \frac{2}{\pi}\arctan\left(\frac{|\boldsymbol{u}_{\mathrm{b}}|^2}{v_{\mathrm{ref}}^2}\right), \tag{10}$$

where $\boldsymbol{u}_{\mathrm{b}}$ is the basal SIA velocity as in Eq. (2). Thus, in the HS-2b the SStA solution is not a replacement of a sliding law. The latter is rather used to determine when and where the corresponding SStA contribution should partly or completely replace it.

The third approach (henceforth HS-3) simply adds up the non-sliding SIA and SStA solutions:

$$\boldsymbol{U} = \boldsymbol{u} + \boldsymbol{v} \tag{11}$$

This superposition of approximations has been employed in recent studies using non-sliding SIA models complemented with a SStA solution as a sliding law (e.g., Winkelmann et al., 2011). It bypasses the need for additional free parameters, such as $r_{\mathrm{thr}}$ and $v_{\mathrm{ref}}$ in the HS-1 and HS-2, respectively. This approach is based on the assumption that on ice shelves the SIA contribution is negligible due to low surface gradients, and therefore the modelled ice flow is dominated by the SStA solution. In the continental interior, the sectors of the ice sheet where basal drag is high enough for vertical shear to dominate are

controlled by the SIA solution (Winkelmann et al., 2011). Since the SIA and SStA solutions are computed over the entire domain, their superposition enables a smooth transition across different flow regimes, ranging from slow ice motion in the interior to a characteristic fast flow of ice shelves, thereby allowing for stress transmission across the grounding line. As in the HS-2, an identification of fast flowing zones is purely diagnostic and not required during the computation of $\boldsymbol{U}$.

## 2.3   Inversion of basal sliding coefficients

We have implemented an iterative method following Pollard and DeConto (2012b) in order to infer the distribution of sliding coefficients $C_0$ that minimises the difference between the modelled and observed ice thickness. The method starts from a



spatially uniform guess value for the distribution of $C_0$ and runs the model forward in time, as described in Section 3. At a given time step, $\Delta t_{\mathrm{inv}}$, the method uses the basal temperature below the pressure-melting point, $T_{\mathrm{m}}$, to identify grounded grid points where basal sliding may occur. If the absolute value of $T_{\mathrm{m}}$ is smaller than the parameter $\gamma$ from Eq. (3), i.e., close to the pressure-melting point, the method computes the difference between the modelled and observed ice thickness, which is then

used to locally adjust $C_0$ at this grid point according to

$$C_0^* = C_0 10^{\Delta H}, \tag{12}$$

where $C_0^*$ is an updated sliding coefficient and $\Delta H = (H - H_{\mathrm{obs}})/H_{\mathrm{inv}}$. Here, $\Delta H$ is the difference between the modelled and observed ice thickness, scaled by a factor, $H_{\mathrm{inv}}$, in order to prevent overshoots. For the same reason and following the implementation by Pollard and DeConto (2012b), variations in the value of the multiplicative factor $10^{\Delta H}$ are further limited

by a range of $\sim .03$ to 30. The parameter $\gamma$ is set to $3\,\mathrm{K}$, in contrast to previous studies using SICOPOLIS where $\gamma = 1\,\mathrm{K}$. Based on our initial tests, this choice provides an easier activation of the inversion procedure, slightly improving the results.

Studies using this iterative technique and other independent inversion methods have shown that potential distributions of sliding coefficients $C_0$ are highly heterogeneous, with values spanning several orders of magnitude (e.g., Pollard and DeConto, 2012b; Arthern et al., 2015). Similarly, and to ensure numerical stability, we limit our inferred values to a range of 1 to

$10^5\,\mathrm{m\,yr^{-1}\,Pa^{-1}}$ during the calibration procedure. Additionally, we implemented the following condition: When the computed surface ice speed reaches an ancillary speed limit at a certain grid point, the local adjustment is halted. This keeps the inversion from over-adjustment of the coefficients when the speed limit has been reached and no noticeable changes occur in response to further adjustments of $C_0$. This additional constraint is applied in order to ensure the numerical stability and keep the modelled ice velocities within the range of observations. For the experiments presented in this study, the lower speed limit is defined as

$0.1\,\mathrm{m\,a^{-1}}$, whereas the upper limit is set to $4000\,\mathrm{m\,a^{-1}}$. This values are based on the observed surface velocities of Rignot et al. (2011).

Our adaptation of the iterative technique involves an additional limiting condition that takes into account temporal variations in the ice thickness. In each grid point, if the adjustment implemented at the previous time step reduces the difference between the modelled and observed ice thickness, further adjustments are suspended. The process is reactivated when the time derivative

of the ice thickness becomes zero or the misfit starts increasing. Our experiments have shown that this additional feature improves the overshoot prevention and enables the use of a smaller $\Delta t_{\mathrm{inv}}$, because further adjustments will only be applied when and where strictly necessary. A further benefit is that it indirectly lets non-local adjustments of $C_0$ to influence the local ice dynamics: If an adjustment applied at the previous time step in the vicinity of a given grid point causes the local misfit to decrease, further local adjustments will still be halted, and vice versa.

## 3 Experimental setup

The modified version of SICOPOLIS described in Section 2 is applied to the entire AIS and fringing ice shelves. The experiments performed during the calibration of the ice sheet-shelf system have their aim to quantify the differences and similarities





between the hybrid approaches. Default values for the parameters controlling the hybrid schemes are $r_{\text{thr}} = 0.5$ (HS-1), and $v_{\text{ref}} = 100 \, \text{m a}^{-1}$ (HS-2a and HS-2b). As mentioned in Section 2.2, the HS-3 does not include any free parameters. Additionally, and for comparison purposes, the same experiments are performed using a SIA-only scheme (henceforth SoS).

The calibration procedure takes an advantage of the improving quality of the modern, continental-scale Antarctic data sets,
such as climatic forcing (Van Wessem et al., 2014), topography (Fretwell et al., 2013), and surface velocities (Rignot et al., 2011). The forcing data serve as time-invariant boundary conditions for our model simulations, which run until a thermal and dynamic quasi-equilibrium is reached. It should be noted that the modern AIS is not necessarily in steady state, and that a transient simulation of, e.g., the entire last glacial cycle would provide a more realistic scenario for the calibration procedure. However, existing reconstructions of the Antarctic paleo-climate and past ice sheet configurations still contain large
uncertainties, with in-situ data being scattered in both space and time. Keeping this in mind, we believe that using modern data sets and equilibrium conditions is a valuable first-order approximation to a more complex model calibration. Furthermore, such equilibrium setup can serve as an initial guess for transient deglaciation simulations, which include time-dependent processes not considered here.

Initial modern conditions for surface topography, ice shelf thickness, and bedrock elevations relative to the present-day sea
level are derived from the BEDMAP2 data set (Fretwell et al., 2013). At the base of the thermal bedrock, geothermal heat flux is prescribed according to the study of Shapiro and Ritzwoller (2004). Boundary conditions at the surface include observational and model-based Antarctic accumulation rates and near-surface air temperature data from the regional climate model RACMO (Lenaerts et al., 2012; Van Wessem et al., 2014), averaged over the period of 1979 to 2010. A simple lapse-rate correction of $0.008 \, ^\circ\text{C m}^{-1}$ accounts for changes in surface elevation. Surface melt is computed by a positive degree-day (PDD) scheme
(Reeh, 1991; Calov and Greve, 2005) using the factors $\alpha_{\text{ice}} = 8 \, \text{mm d}^{-1} \, ^\circ\text{C}^{-1}$ and $\alpha_{\text{snow}} = 3 \, \text{mm d}^{-1} \, ^\circ\text{C}^{-1}$ (ice equivalent) for ice and snow, respectively (Ritz et al., 2001), and a standard deviation of $\alpha_{\text{std}} = 5 \, ^\circ\text{C}$ for the statistical fluctuations of air temperature.

The bulk of the calibration consists of the inversion technique for the distribution of sliding coefficients $C_0$ (Section 2.3) that exerts a dominant control on the resulting ice distribution and its fit to the observational data. The inversion procedure starts
from a homogeneous guess value of $C_0 = 1 \, \text{m yr}^{-1} \, \text{Pa}^{-1}$, which is equal to the lower limit. The time step between adjustments and the scaling factor in Eq. (12) are set to $\Delta t_{\text{inv}} = 50$ years and $H_{\text{inv}} = 5000$ metres, respectively. We follow the method by Pollard and DeConto (2012b) and allow for a free evolution of both the ice sheet and ice shelf thickness, but their interface (the grounding line) is kept at its present-day observed position. Free evolution is needed because the inversion of the sliding coefficients requires an evolving ice thickness that will be routinely compared to observations. The reasons for a constrained
grounding line are twofold: Such approach 1) ensures a one-to-one comparison with observations during the calibration and 2) prevents artificial transitions between grounded and floating areas caused by equally artificial effects of the unrealistic initial thermal regimes and $C_0$ distributions that evolve from the initial guess values. Our tests show that such artefacts can produce feedbacks that are difficult or impossible to reverse. For the same reason, glacial isostatic adjustment is not accounted for, and ice shelf fronts are constrained to the observed locations. As mentioned above, the ice shelf thickness is allowed to evolve, but
basal melt rates are adjusted at each time step in order to keep the modelled ice shelves as close as possible to observations.





This ensures a consistent computation of mass fluxes across all flow regimes, and does not overlap with the inversion of sliding coefficients, since these are not applied in floating ice sectors.

In this study, the calibration procedure is divided into two steps. First, a relaxation scheme is applied to the evolution of the modelled ice thickness, $H$. Here, the difference between the current solution of the ice thickness equations, $H_{\mathrm{new}}$, and the solution from the previous time step, $H_{\mathrm{old}}$, is scaled at every time step by a factor $h_{\mathrm{rlx}}$ ranging between 0 and 1, as following:

$$H = H_{\mathrm{old}} + h_{\mathrm{rlx}}(H_{\mathrm{new}} - H_{\mathrm{old}}). \tag{13}$$

For a time-invariant forcing, our tests have shown that different values of $h_{\mathrm{rlx}}$ will result in very similar equilibrium states. The relaxation simply delays the time point at which this state is reached. However, such relaxation procedure allows for bigger time steps for the computation of the topography evolution, without affecting the internal temperature evolution. In this way, an equilibrium with the boundary conditions will be reached faster by the temperature field than by the topography. This effectively minimises transient effects in the closely associated ice thickness and velocity fields, especially at the beginning of the simulations when model parameters follow the initial guess values. More importantly, it allows us to simultaneously apply the inverse technique described in Section 2.3, in contrast to approaches in which the topography is fixed. This ensures that the modelled ice thickness distribution stays as close as possible to the initial observed distribution. The value of the relaxation factor is set to $h_{\mathrm{rlx}} = 0.001$ throughout this first stage of the calibration driven over a simulation time of 100 thousands of years (kyr), using a time step of 10 years. The second stage of the calibration is activated once the thermal equilibrium has been reached. It deactivates the relaxation and applies a smaller time step of 1 year over a simulation time of $200\,\mathrm{kyr}$, which provides a time-span long enough to attain a dynamic equilibrium.

All input fields are projected onto a regular, rectangular, polar stereographic grid covering the entire Antarctic continent and the surrounding Southern Ocean, with a nominal horizontal resolution of $40\,\mathrm{km}$, corresponding to $151 \times 151$ grid points. Our choice of a rather coarse resolution has been motivated by the large number of experiments presented in Section 4, spanning hundreds of thousands of years, and the fact that Pollard and DeConto (2012b) have shown that the results remain essentially unchanged when the horizontal resolution is increased to 20 and $10\,\mathrm{km}$ (the latter in a regional, nested simulation), even in rapidly flowing sectors. Moreover, fixing the grounding line at its observed position prevents unrealistic migrations that would otherwise arise from the use of low resolution. However, our tests using an increasingly higher resolution have identified areas where the modelled ice flow is more sensitive to the resolution used. These mainly occur close to the ice sheet margins where a glacier flux gate is often represented by only one grid cell at low resolution. In such regions the use of a finer grid allows for a more detailed treatment of the topographically constrained glacial flow. In the context of this comparison study, however, these limitations equally affect the performance of all hybrid schemes and do not impact our conclusions. In the vertical direction, ice columns consist of 91 layers (11 equidistant grid points for temperate ice and 81 grid points for "cold" ice densifying towards the base, which overlap at their interface), mapped to a $[0,1]$ interval using a sigma transformation (Greve and Blatter, 2009).



## 4 Results and discussion

In this chapter we present an ensemble of simulations of the AIS that aim to comprehensively evaluate and get insight into different hybrid schemes combining the SIA and SStA. Our evaluation uses the degree of agreement between the modelled and observed ice sheet geometries and flow patterns as a measure of their performance, allowing for a point-by-point quantification
of the model errors.

Keeping in mind that each hybrid scheme builds upon the SIA solution, partially or entirely replacing it at variable locations with the SStA solution, we have also included the results from the SIA-only scheme in our comparison. In particular, this enables a qualitative separation of relative contributions of the SIA and SStA, providing new insight into the internal differences between the schemes and their applicability to ice sheet areas with diverse dynamical characteristics.

By applying an automated model calibration against the observed ice thickness to each of the hybrid schemes we infer spatial distributions of poorly constrained sliding coefficients as a proxy for mechanical conditions at the ice-bedrock interface and assess their sensitivity to the choice of a particular hybrid scheme. In addition, the influence of variations in parameters controlling the internal operation of the hybrid schemes is assessed for a wide range of parameter values.

### 4.1 Comparison of equilibrium states

As described in Section 3, our experiments use the BEDMAP2 observational data set (Fretwell et al., 2013) as an initial ice sheet configuration, running the model forward in time under a relaxation scheme (Eq. (13)) until the temperature distribution within the ice sheet reaches a quasi-equilibrium. After this initialisation, the relaxation is deactivated and the model runs under an automated calibration procedure (Section 2.3) until a full thermal and dynamic equilibrium is attained. This equilibrium is defined as the point in time in which the variation of total grounded ice volume becomes negligible.

Starting from the initialised states, the time required to reach an equilibrium varies from scheme to scheme (Fig. 1b). The HS-1 attains a virtually invariable state after only $50\,\mathrm{kyr}$, as opposed to the HS-2a that requires about $150\,\mathrm{kyr}$. The length of calibration for other schemes falls within this range. Compared to the SoS, the computation of SStA velocities in grounded ice sectors implies an extra computational effort, with the computing time increasing by a factor of $\sim 4$ for the hybrid schemes. The computing time of the HS-1 is somewhat shorter due to the prognostic identification of ice streams that prevents the
computation of SStA velocities over the entire ice sheet. However, we have observed that iterative solvers in the model require a substantially smaller number of iterations when the hybrid schemes are used, making them numerically more stable compared to the SoS.

The calibration procedure applied to all schemes yields total grounded ice volumes which are in a close agreement with the reference value of $2.55 \times 10^7\,\mathrm{km}^3$ from the observational data (Fretwell et al., 2013), with the maximum deviation being
below $2\,\%$. Individual values of the modelled ice volumes and their respective deviations for most experiments are summarised in Table 2. The best fit to observations is obtained using the HS-2b, which corresponds to an underestimation of the total grounded ice volume by less than $0.3\,\%$. The other schemes produce relatively larger misfits, with the HS-2a simulation yielding the greatest deviation of $1.88\,\%$ arising from an overestimation of the total ice volume. It can be observed that the





schemes that include basal sliding in the computation of the SIA solution (HS-1, HS-2b, and SoS) tend to produce smaller ice sheets than those using the SStA as a sliding law (HS-2a and HS-3). The smallest ice sheet is produced by the SoS that underestimates the total grounded ice volume by $1.55\%$. The evolution of the total grounded ice volume depicted in Fig. 1b also shows different types of signals: The SoS and HS-1 solutions contain high-frequency variations even after an equilibrium
is reached. These are contrasted by the HS-2b solution that displays low-frequency oscillations and the smooth, noise-free curves obtained from the HS-2a and HS-3 schemes.

Figure 1a shows total ice thickness errors for all schemes. The errors are computed as an average of the absolute values of the misfit in all grounded grid points and remain below $60$ metres for all hybrid schemes (Table 2). Among these, the largest error of $59.8\,\mathrm{m}$ is produced by the HS-2a, while the smallest error of $49.9\,\mathrm{m}$ is obtained using the HS-2b. For the former, an
$83\%$ of the misfit is due to overestimations of ice thickness, while the latter shows a nearly even split between under- and overestimations. This is in accordance with the results shown in Fig. 1b, where the misfits obtained from the two schemes using the SStA as a sliding law are dominated by an excessive ice thickness. As mentioned above, the opposite is generally not the case for the schemes that include a sliding component within the SIA. For example, only a quarter of the misfit produced by the HS-1 can be attributed to areas with ice thickness deficit. In comparison, the total error produced by the SoS is $73.1\,\mathrm{m}$,
with $72\%$ of the misfit coming from the underestimation of ice thickness.

The results depicted in Fig. 1 reflect only a generalised, time-dependent information that can be inferred from each run. In order to have a precise overview of the results, Fig. 2 (left column) shows the corresponding spatial maps of ice thickness errors. All schemes provide a reasonably good fit to the observational data in the continental interior, with larger discrepancies mainly occurring at the ice sheet margins. It can be readily observed that the modelled ice sheet is too thick over the mountainous
regions and in the region between the Shakleton Range and the Pensacola Mountains, which according to observations is not characterised by steep bedrock topography gradients. These are common features for all schemes, independently of the approach chosen for the sliding component. This can be explained by frozen conditions at the ice base, which prevent the initiation of the basal sliding, thereby restricting the influence of the calibration procedure. The distribution of zones where basal temperatures are far from the temperature at the pressure-melting point are depicted as the white-coloured areas in Fig. 2
(bottom row). In general, these areas coincide with the locations where the largest ice thickness errors occur. On the other hand, underestimations of the ice thickness are sparsely distributed at and around the ice margins. In these areas, however, sliding is identified and the inversion of $C_0$ is performed. The SoS produces too thin ice along most of the ice sheet margins, which explains the high percentage of the ice thickness error arising from underestimations (Fig. 1a). In contrast, the hybrid schemes produce error patterns that differ only slightly between each other. The only exception is the HS-2b, which resembles
the patterns from the SoS, albeit exhibiting smaller underestimations at the ice sheet margins.

Furthermore, we evaluate the performance of the inversion procedure over the areas where it is applied directly. For this purpose we have calculated averaged absolute errors in the ice thickness across regions where basal sliding operates (Fig. 1a). It should be kept in mind that the calibration procedure also affects the ice masses located in the immediate proximity to sliding areas through surface elevation gradients and/or stress transmission. Mean ice thickness errors over the sliding areas
are smaller than those estimated for the whole ice sheet, although the degree of relative improvement varies from scheme to





scheme (Table 2). For example, the misfit produced by the SoS decreases only by $\sim 10\%$ if calculated over the sliding ice sheet sectors, while more than $50\%$ of the errors resulting from the HS-2a occur over the areas where $C_0$ is not inverted. For all hybrid schemes, the percentage of the errors associated with an underestimation of the ice thickness increases substantially, supporting the observation that the modelled ice is excessively thick mainly in the regions where the calibration procedure does

not operate.

The inferred distributions of $C_0$ are shown in Fig. 2 (right column). In general, the areas where the inversion is performed are similar for all schemes, although there is a significant spread in the inverted values. Conditions favourable for sliding are predicted over more than a half of the ice-covered area (Table 2). The HS-1 scheme predicts a minimum corresponding to $53\%$ of the total grounded ice area, contrasted by the corresponding percentages for the other schemes ranging between $60\%$

and $66\%$. The upper limit of $10^5 \, \mathrm{m\,yr^{-1}\,Pa^{-1}}$ for the sliding coefficient is reached by all schemes across up to $5\%$ of the ice-covered land, with their highest concentration occurring in the Siple Coast region, where ice streams flow rapidly over a smooth and deformable bed provided by strong lubrication from water saturated subglacial sediments (e.g., Blankenship et al., 1986; Alley et al., 1987; Kamb, 2001). The upper limit is also reached in the Coats, MacRobertson, and Ellsworth Lands. The $C_0$ values at the lower limit of $1 \, \mathrm{m\,yr^{-1}\,Pa^{-1}}$ are also present in the estimates from all runs, with a relatively higher coverage

ranging between $\sim 8\%$ and $\sim 30\%$ of the ice sheet area. The SoS and HS-2b infer this value throughout most of the West AIS and over vast parts of the East AIS, particularly at the ice margins. Its incidence is relatively lower for the other hybrid schemes, especially for the HS-1 and HS-2a. Quantitatively, the good agreement between the inferred coefficient distributions is mainly restricted to the areas where $C_0$ reaches an upper or lower limit. In other areas the values generally vary across orders of magnitude.

Direct comparison of our results to those from Pollard and DeConto (2012b) is hindered by the differences in the sliding laws and hybrid schemes. Nevertheless, we have also found a good qualitative agreement in the inferred distributions of $C_0$, with similar patterns of high vs. low values. For instance, both studies identify the Siple Coast and Coats Land regions as areas where the ice streaming flow is driven by basal conditions favourable for sliding. A similar agreement is found for the Thwaites and Pine Island Glacier areas, as well as in the MacRobertson Land. Low values of $C_0$ are predicted in the continental

interior of West Antarctica and over most of the East AIS. Despite the differences between our model and that of Pollard and DeConto (2012b) we expect that significant modifications to their hybrid scheme would also result in similar perturbations in the inverted values of $C_0$.

## 4.2   Analysis of the SIA and SStA contributions

As described in Section 2.2, different hybrid approaches are not expected to produce exactly the same equilibrated ice velocity

fields. We demonstrate this in Fig. 3, where the modelled steady-state surface velocities derived from the SoS and hybrid schemes are compared to a continental-scale observational data set (Rignot et al., 2011). This high-resolution ($900 \, \mathrm{m}$) data set contains many small-scale features that are barely resolved by the model, due to its lower resolution. Nevertheless, all schemes are able to reproduce the observed range of ice flow regimes, distinguishing between ice sheet areas with small velocities near ice divides, and high velocity flanks around the ice margins. In the transition zones between the continental interior and the ice

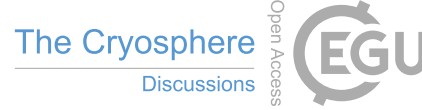



margins, all schemes reproduce to some extent the fast flowing ice streams identified by observations. However, in contrast with the results from the hybrid schemes, surface ice velocities derived from the SoS are contaminated by noise-like patterns, which are especially visible in the areas of rapid ice flow. We attribute these artefacts to a combination of lacking stress transmission in the SIA, which allows for steep gradients in the modelled velocities, and the calibration procedure, which can potentially

amplify these gradients through local adjustments of $C_0$.

Although the overall character of the observed surface ice velocities is qualitatively well reproduced by all hybrid schemes, modelled ice flow is clearly too fast at and around several ice stream locations, such as, for example, in the Siple Coast. Furthermore, modelled surface velocities are generally overestimated at the flux gates of most outlet glaciers, in line with the resolution-related limitations mentioned in Section 3. These overestimations are particularly large in the SoS simulation, even

though the discrepancies between the modelled and observed ice thickness are small (Fig. 2, left column), owing to the values of $C_0$ inferred from the calibration procedure (Fig. 2, right column). On the other hand, overestimations of the ice flow velocity by the hybrid schemes at the flux gates are smaller than those derived from the SoS simulation, but they cover a wider area and reach further upstream. Similar observations have been made by Pollard and DeConto (2012b) using a different hybrid ice sheet-shelf model and a different set of topographic ice observations and external forcing. They proposed that the inferred

overestimation of surface velocities in these areas may be caused by a coarse model resolution, exaggerated snowfall rates or an excessive internal deformation compared to sliding near the ice margins.

In order to quantitatively evaluate the model fit to observations, we have calculated point-by-point ratios between the modelled and observed surface velocities (Fig. 3). As mentioned above, the modelled velocities are in a good agreement with observations in the continental interior characterised by a slow ice motion. In contrast, at the margins the ice flow speed pre-

dicted by the hybrid schemes sometimes reaches values that are several hundred times higher than in the observational data set, but this mostly happens across areas where our model generates a non-existent fast flow. Arguably, one of the best examples of such model artefacts is the former Ice Stream C in the Siple Coast, which has been stagnated for $\sim 150$ years (Hulbe and Fahnestock, 2007; Engelhardt and Kamb, 2013). In some cases, the locations of the modelled ice streams are shifted relative to the observed ones, thereby generating adjacent zones of under- and overestimations of surface velocity. In other cases, rapid ice

flow deviates to either side and is pushed to merge with adjacent ice streams. These shifts may originate from local deficiencies in the bedrock topography data accentuated by its projection onto a coarse horizontal grid.

The mean errors in the absolute surface ice velocity fall within the range of $\sim 30$ to $\sim 80\,\mathrm{m\,yr^{-1}}$ (Table 2), with the HS-2b and SoS producing the minimum and maximum misfits, respectively, analogously to the results for the mean ice thickness error discussed in Section 4.1. In the SoS simulation, a general underestimation of the ice thickness near the ice sheet margins

coincides with areas where sliding coefficients tend to reach the lower limit prescribed for the inversion. In turn, the use of such low values triggers a slowdown of the ice flow in the modelled surface velocities in the transition zone. Although the results of the HS-2b simulation presented in Section 4.1 are in many aspects similar to those from the SoS, their respective skills in reproducing observations are at the opposite ends. Equation 10 uses the basal velocity from the SIA to compute the weights of relative contributions of the shallow approximations, thereby enabling the SStA contribution where sliding velocities from

the SIA are high. In these rapidly flowing sectors, we attribute the better performance of the HS-2b compared to the SoS to the





inclusion of the stress transmission by the SStA. It also fosters a SStA-dominated modelled ice flow in the surroundings of ice streams, particularly upstream, thereby improving the overall fit to observations.

As described in Section 2.2, the hybrid schemes used in this study mainly differ in how the SStA weight, $w$, is computed. This does not only imply the relative contributions from the shallow approximations, but also the locations where such combination

is implemented. In order to provide a deeper insight into the internal operation of the hybrid schemes, Fig. 4 compares the equilibrated distributions of $w$. It can be immediately observed that the distributions of $w$ produced by the hybrid schemes are very different, both in the spatial coverage and individual contributions of the SIA and SStA. For example, the percentage of the grounded ice area where the modelled ice flow is dominated by the SStA ($w > .75$) is $\sim 30\,\%$ for the HS-1, but only $\sim 3\,\%$ and $\sim 8\,\%$ for the HS-2a and HS-2b, respectively (Table 2). Furthermore, the transition between the SIA- and SStA-dominated

regimes of the modelled ice flow appears patchy in the HS-1, whereas it is smooth and collocated with the present-day ice streams in the HS-2a, thereby resembling the velocity field from the observational data set. The transitions in the HS-2b are sharp, implying a simple differentiation between fast and slow ice flow areas. This particular scheme exhibits a SIA-dominated ice flow regime ($w < .25$) over $81\,\%$ of the grounded ice area that may help explain the high degree of similarity with the SoS, especially keeping in mind that it also includes basal sliding in the SIA. In contrast, the HS-2a uses a non-sliding SIA, but still

produces a similar percentage of ice-sheet–covered area dominated by the SIA. In these sectors, differences in the ice thickness derived from the HS-2a and HS-2b (Fig. 2, left column) can be attributed to the presence or absence of the basal sliding in the SIA that tends to prevent overestimations, e.g., in interior East Antarctica, and causes underestimations in some other sectors, such as, for example, the surroundings of Dronning Maud and MacRobertson Lands.

The disparity in the ways how different hybrid schemes measure relative SIA and SStA contributions may explain the nature

of the misfits between the modelled ice flow and observations. This can be visualized using scatter plots of the modelled vs. observed surface velocities for each scheme, color-coded for the values of the corresponding $w$ distributions (Fig. 4). In general, underestimations of the modelled velocities occur across slowly flowing ($< 10\,\mathrm{m\,yr^{-1}}$) areas, which are dominated by the SIA, while the areas of fast flow dominated by the SStA are responsible for most of the overestimation. At first glance, it may seem that overestimations are caused by an excessive contribution of the SStA, but a cursory comparison with the SoS

scatter plot shows that this is not necessarily the case: The largest overestimations occur in the SoS simulation, with surface velocities reaching the upper permitted limit clustered in the upper part of the scatter plot (Fig. 4). As mentioned above, the SoS and HS-2b share many similarities, and a comparison between their respective scatter plots shows that the use of the hybrid scheme reduces both underestimations in the lower velocity range and overestimations in the fast flowing areas.

Here we attempt to isolate the influence of relative SIA and SStA contributions on different flow regimes, which can be

characterised by weights between $w = 0$ (slow, SIA-dominated flow) and $w = 1$ (fast, SStA-dominated flow). For different intervals within this range, we computed averaged errors arising from under- and overestimation of the surface velocity (Fig. 5). As has been demonstrated in Fig. 4, overestimations in the modelled velocities escalate for all schemes as $w$ approaches 1. It is readily observed that the departures of the modelled surface velocities produced by the SoS simulation (which discounts the SStA contribution) from observations are larger than those from the hybrid schemes, in terms of both under- and overestima-

tions and for all types of flow regimes. This supports our assertion that the inferred overestimations in the fast flowing regions





are not necessarily caused by an excessive contribution from the SStA solution. Overall, the performance of different hybrid schemes is quantitatively similar, with no striking outliers occurring when they are validated against observational data. The HS-1 shows larger velocity misfits across the slow ice sheet sectors if evaluated against the distributions of $w$ from the HS-2a and HS-2b. This rise in velocity misfits is caused by an increased use of the SStA by the HS-1 across the areas where the HS-2a

and HS-2b favour a SIA-dominated ice flow (Fig. 4).

### 4.3   Intercomparison of the inferred basal sliding coefficients

Although the performance of different hybrid schemes is quantitatively similar when evaluated against observations, the inferred values of basal sliding coefficients ($C_0$) vary by orders of magnitude in many regions of Antarctica (Fig. 2, right column). We quantify this variability by looking at the standard deviation of the distributions of $C_0$ (Fig. 6). Approximately 35 % of the

total area where the calibration procedure is activated is characterised by a standard deviation of $100\,\mathrm{m\,yr^{-1}\,Pa^{-1}}$ or higher. At the other end, only a quarter of the sliding area displays a standard deviation of $1\,\mathrm{m\,yr^{-1}\,Pa^{-1}}$ or less. Of this latter fraction, almost two thirds occur across areas where the adjusted value of $C_0$ reaches either the lower or the upper limit prescribed for the inversion procedure. This observation suggests a possibility of even higher variability in the distribution of $C_0$ if, for example, the upper limit were to increase. It is important to keep in mind that these large differences between the inferred distributions

of $C_0$ arise solely from the differences in the hybrid schemes, namely in their ways to combine the shallow approximations, since all other model components are exactly the same for all experiments.

   To demonstrate the effects arising from the variability in the inferred distributions of $C_0$, we performed additional experiments, which start from the equilibrium states described in Section 4.1. In these experiments, we exchange the distributions of $C_0$ inferred from the HS-2a and HS-2b, and then run the model over the period of $100\,\mathrm{kyr}$. As described in Section 2.2,

these schemes slightly differ in how they compute the SStA contribution, mainly due to different techniques used to account for basal sliding. Although both schemes identify similar locations of rapidly flowing ice (Section 4.2) their inferred distributions of basal sliding coefficients contain the highest variability among all hybrid schemes implemented in this study. At the end of the additional 100kyr runs, the mean misfit between the modelled and observed ice thickness is above $200\,\mathrm{m}$ for the HS-2a, and above $300\,\mathrm{m}$ for the HS-2b (not shown). This represents increments of $\sim 250\,\%$ and $\sim 550\,\%$ in the deviations from

observations, respectively.

   We generalize these experiments and further exemplify the significance of the associated uncertainties in the retrieved basal sliding parameters by prescribing a median of the inferred distributions of $C_0$, computed from all hybrid schemes, at the base of the ice sheet. Our choice of a median over an average is motivated by our initial tests in which generally larger values of $C_0$ inferred from the HS-2a tend to produce an average biased towards this particular distribution. Figure 7 shows differences

between the modelled and observed ice thickness at the end of these additional runs. Comparison with Fig. 2 reveals a general degradation of the fit between the model and observations. The HS-2a and HS-2b exhibit the largest sensitivity to the change in the basal sliding parameters, with a significant amplification of over- and underestimations of the ice thickness occurring across most of the ice sheet, respectively. For the HS-2a, an absolute ice thickness misfit increases by $\sim 150\,\%$ to a mean value of $150\,\mathrm{m}$ (Table 3), of which 97 % is due to overestimations. On the other hand, the misfit increases by $\sim 230\,\%$ to a mean





value of $172\,\mathrm{m}$ in the HS-2b simulation, with underestimations of the ice thickness accounting for $89\,\%$ of the total difference. This degree of degradation is less pronounced when the HS-1 and HS-3 are used. The misfit from the former increases by only $\sim 30\,\%$, displaying a mixture of areas where the modelled ice thickness is either too large or too small relative to observations. The latter shows an intermediate degree of sensitivity to a change in basal parameters and exhibits similar misfit patterns as the

HS-2b, albeit the magnitude of the underestimation is smaller and the mean absolute error in the ice thickness remains below $100\,\mathrm{m}$.

### 4.4 Exploration of the hybrid parameter space

Disparity in the results from the hybrid schemes presented in the previous sections showcases the impacts of slight changes in the model representation of fast flowing zones of ice sheets, even though it corresponds to a small fraction of the parameter

space in ice-sheet models. Within this parameter space, the use of somewhat arbitrary threshold and reference quantities by some of the hybrid schemes allows us to perform an additional series of experiments in which we explore the sensitivity of the results to variations in these parameters.

As described in Section 2.2, the hybrid schemes HS-1, HS-2a, and HS-2b use the weight $w$ to calculate relative contributions of the SIA and SStA. For the HS-1, the computation of $w$ involves a prescribed threshold for the slip ratio of grounded ice

(Eq. (7)), which determines the locations where the hybrid velocity is calculated. The experiments discussed in the previous sections use a default value of $r_{\mathrm{thr}} = 0.5$, meaning that the SIA and SStA solutions are combined only in areas where the basal velocity is at least half of the surface velocity. In contrast, the HS-2a and HS-2b combine both shallow approximations everywhere, and $w$ is computed using a prescribed reference velocity (Eqs. 9 and 10), using a default value of $v_{\mathrm{ref}} = 100\,\mathrm{m\,yr}^{-1}$. If the SStA velocity in the HS-2a or the basal SIA velocity in the HS-2b reaches this reference value, the schemes assign equal

weights to the SIA and the SStA. In general, higher values of $r_{\mathrm{thr}}$ or $v_{\mathrm{ref}}$ imply less contribution of the SStA solution, and vice versa.

The effects of variations in these parameters on the evolution of the total grounded ice volume during the calibration procedure are demonstrated in Fig. 8. Here we test values within the range of what can be considered lower and upper limits for each hybrid scheme.

For the HS-1, the upper limit for the slip ratio threshold, $r_{\mathrm{thr}} = 1$, implies the use of a SIA-only solution, whereas the lower limit, $r_{\mathrm{thr}} = 0$, implies that a combination of SIA and SStA solutions is applied everywhere, determined by the weight $w$ computed using the slip ratio. Within the range of tested values, maximum deviations from the observed grounded ice volume are below $1.5\,\%$. Only the use of $r_{\mathrm{thr}} = 1$ produces an ice sheet that is smaller than observed. An analysis of the mean absolute difference between the modelled and observed ice thickness (Fig. 9) reveals that the use of values below $r_{\mathrm{thr}} = 0.5$ leads to a

larger misfit with observations, which is also the case for the upper limit, $r_{\mathrm{thr}} = 1$. However, $r_{\mathrm{thr}} = 0.75$ produces a slightly improved result, suggesting that at some higher value the fit will start to decrease.

In the HS-2a, all tested values produce an ice sheet that is larger than observations. This is caused by a general overestimation of the ice thickness at the continental interior. Here, the modelled ice flow is slow and dominated by the non-sliding SIA. Thus, the weights $w$ are small, implying a negligible contribution of the SStA, independently of the value of $v_{\mathrm{ref}}$. Values smaller





than $v_\mathrm{ref} = 100\,\mathrm{m\,yr}^{-1}$ are found to produce total grounded ice volumes and mean ice thickness misfits that are close to the results of the reference run ($v_\mathrm{ref} = 100\,\mathrm{m\,yr}^{-1}$), although with a slight improvement of the model fit to observations. Values of $v_\mathrm{ref} > 100\,\mathrm{m\,yr}^{-1}$ lead to a larger misfit, with the highest tested value of $v_\mathrm{ref} = 1000\,\mathrm{m\,yr}^{-1}$ producing a grounded ice volume that deviates by $2.5\,\%$ from observations. The use of even larger values of $v_\mathrm{ref}$ asymptotically decreases the contribution of the

SStA. It is important to note, however, that since the HS-2a does not include sliding in the SIA component, its solutions will never approach that of the SoS.

In contrast, the HS-2b activates the sliding law even in the regions where ice streams have not been identified by the scheme, as long as the conditions for sliding described in Section 2.1 are fulfilled. This may explain the similarity between the grounded ice volumes produced by the SoS and the HS-2b solutions, using a reference velocity set to the upper limit of

$v_\mathrm{ref} = 1000\,\mathrm{m\,yr}^{-1}$. This parameter value produces a deviation of $2\,\%$ from the observed ice volume, similarly to the misfit obtained from the simulating using the lower limit of $v_\mathrm{ref} = 5\,\mathrm{m\,yr}^{-1}$ tested in this study. For values higher than the default value of $v_\mathrm{ref} = 100\,\mathrm{m\,yr}^{-1}$, the simulations produce an ice sheet that is smaller than observed. The use of $v_\mathrm{ref} = 60\,\mathrm{m\,yr}^{-1}$ leads to the best fit between the modelled and observed ice volumes. Here, the misfit of $0.12\,\%$ in the total ice volume is accompanied by the the second smallest misfit in ice thickness among all schemes and parameter values, reaching $\sim 49\,\mathrm{m}$. The

smallest misfit overall is produced by the HS-2b, using a reference velocity of $v_\mathrm{ref} = 30\,\mathrm{m\,yr}^{-1}$, although here the deviation of $0.66\,\%$ in the grounded ice volume is higher.

However, it is important to keep in mind that the calibration procedure has a limited power to reduce the misfits between the model and observations in areas where no sliding occurs. This is why some deviations are expected from our numerical experiments, in particular due to overestimations of the ice volume over mountainous regions where the calibration procedure

does not operate. Therefore, a perfect fit to the observed ice volume obtained by any of the schemes would likely involve underestimations of ice thickness in other regions.

## 5   Conclusions

We implemented and compared the performance of four hybrid schemes for the combination of the Shallow Ice and Shelfy Stream Approximations into the ice-sheet model SICOPOLIS. The use of shallow approximations enables, continental-scale,

long-term paleo-simulations of entire ice sheets, preventing the restrictive computational expenses of more complex models. Moreover, hybrid schemes overcome the limitations of simpler SIA-only models in regions characterised by rapid ice flow driven by basal sliding. The hybrid schemes in this study differ in their ways to: 1) compute relative contributions of the shallow approximations, 2) identify areas where a combination of shallow approximations is applied, and 3) account for basal sliding.

By adapting a simple inverse technique to infer the distribution of basal sliding parameters from observational topographic data sets (Pollard and DeConto, 2012b), we show that all hybrid schemes produce dynamic equilibrium states that are in good agreement with observations. For all schemes, mean ice thickness misfits are below $\sim 80\,\mathrm{m}$ and total grounded ice volume deviations do not exceed $2.5\,\%$ for a wide range of the associated parameter uncertainties. For optimal parameter choices,



mean errors in the ice thickness remain below $50\,\mathrm{m}$. For comparison, present-day simulations in continental-scale ice-sheet models typically produce widespread errors in the ice thickness reaching hundreds of metres, with deviations in total grounded ice volume that can can exceed $10\,\%$ (e.g., de Boer et al., 2015).

We also computed a non-hybrid, SIA-only solution to allow for a qualitative separation of relative contributions of the SIA and SStA. Such direct comparison is possible because all schemes analysed in this study share identical model components. Although the calibration procedure applied to the SIA-only scheme also produces a reasonable fit to observations, the misfits are overall larger than those from hybrid schemes. Moreover, the modelled surface velocities exhibit noise-like patterns at and around locations where rapid ice flow develops. Since these patterns are not present in the hybrid solutions, we attribute their occurrence to the lack of stress transmission in the SIA flow.

We find that individual weights assigned to the SIA and SStA solutions vary significantly from scheme to scheme. For the schemes in which the SIA/SStA weights are computed, the modelled ice flow is dominated by the SIA over $50\,\%$ to $90\,\%$ of the AIS, with the predominant contribution of the SStA generally limited to the locations and surroundings of the observed ice streams. An inclusion of a sliding component in the SIA tends to prevent an excessive accumulation of ice in the continental interior of the ice sheet, as opposed to the scheme that uses the SStA solution as a sliding law but assigns relatively small weights to it across these areas. We have found that this particular scheme tends to increase basal sliding coefficients in an attempt to compensate for an insufficient sliding that would otherwise lead to a larger misfit. The scheme adopted by Winkelmann et al. (2011), which adds up non-sliding SIA and SStA solutions instead of using the SIA/SStA weights, to some extent reduces overestimations of the ice thickness in the continental interior, and thus the values of basal sliding coefficients.

Here we use the fact that large portions of the AIS are dominated by the SIA to compare a hybrid scheme including basal sliding in the SIA solution with the SIA-only scheme. The latter generates a general underestimation of the ice thickness at the ice sheet margins, while the former effectively reduces them. This improvement can be primarily attributed to the inclusion of the stress transmission in the SStA at and around the fast flowing regions, where the assumptions behind the SIA are violated.

Comparison of our results with an independent, observational data set of surface ice velocities reveals a reasonable agreement in the continental interior of Antarctica. However, the modelled ice flow appears to be too fast at the ice sheet margins in all hybrid schemes, especially close to the observed ice stream locations. These misfits between the modelled and observed surface velocities are contrasted by relatively small differences between the modelled and observed ice thickness in these areas. Such misfits might originate from other factors such as, e.g., the neglect of the paleoclimate signal within the AIS implied by the assumption of an equilibrium, the lower model resolution, and deficiencies in the forcing data sets (Pollard and DeConto, 2012b).

Although all hybrid schemes produce a comparably good fit to the observational topographic data set, the inferred values of basal sliding coefficients largely differ between different schemes. Therefore, the discrepancies in the representation of ice dynamics in different hybrid schemes are compensated by a high degree of uncertainty associated with this poorly constrained parameter. Nevertheless, the schemes mostly agree on the areas where the ice base is at or close to the local pressure-melting point. Furthermore, there is a general agreement in the patterns of hard vs. soft basal conditions obtained from each calibration





run. Although any attempt to quantify a local potential for sliding would require new and improved observational data sets on basal conditions, the robustness of the inferred patterns provides an initial guess for their real distribution.

The high variability in the inferred parameters provided an opportunity to quantify the effects of the uncertainty in basal conditions. By prescribing averaged and swapped distributions of basal sliding coefficients derived from different hybrid schemes,

5   we show that they are not exchangeable between the schemes, since their use leads to a strong degradation of the fit between the model and observations. This suggests that results of a model calibration and/or initialisation cannot be straightforwardly transferred to a model that uses a different level of approximation to the Stokes equations. Nevertheless, this paper shows that a simple inverse method for the distribution of $C_0$ can be easily adapted into a variety of modelling approaches. We believe that such implementations can provide the internal consistency required to avoid the aforementioned misfits between the model and

10  observations.

*Acknowledgements.* This work has been funded by the Helmholtz graduate research school GeoSim. RG was supported by Grants-in-Aid for Scientific Research A (Nos. 25241005 and 16H02224) of the Japan Society for the Promotion of Science (JSPS) and by the Arctic Challenge for Sustainability (ArCS) project of the Japanese Ministry of Education, Culture, Sports, Science and Technology (MEXT). The simulations were performed on the GFZ Linux Cluster GLIC. This study utilizes data from the regional atmospheric climate model RACMO2.3.



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





**Table 1.** Symbols and values for the model parameters used in this study.

| Symbol | Description | Units | Value |
|---|---|---|---|
| $g$ | gravitational acceleration | $\mathrm{m\,s^{-2}}$ | 9.81 |
| $\rho$ | density of ice | $\mathrm{kg\,m^{-3}}$ | 910 |
| $T$ | absolute temperature of ice | K | |
| $T_\mathrm{m}$ | temperature below pressure-melting point | K | |
| $\kappa$ | heat conductivity of ice | $\mathrm{W\,m^{-1}\,K^{-1}}$ | $9.828\mathrm{e}^{-0.0057T}$ |
| $c$ | specific heat of ice | $\mathrm{J\,kg^{-1}\,K^{-1}}$ | $146.3 + 7.253T$ |
| $L$ | latent heat of ice | $\mathrm{kJ\,kg^{-1}}$ | 335 |
| $\beta$ | Clausius–Clapeyron gradient | $\mathrm{K\,m^{-1}}$ | $8.7 \times 10^{-4}$ |
| $R$ | universal gas constant | $\mathrm{J\,mol^{-1}\,K^{-1}}$ | 8.314 |
| $\kappa_\mathrm{r}$ | heat conductivity of the lithosphere | $\mathrm{W\,m^{-1}\,K^{-1}}$ | 3 |
| $\alpha_\mathrm{ice}, \alpha_\mathrm{snow}$ | PDD factors for ice and snow | $\mathrm{mm\,d^{-1}\,{}^\circ C^{-1}}$ | 8, 3 |
| $\alpha_\mathrm{std}$ | PDD standard deviation | $^\circ$C | 5 |
| $\sigma_\mathrm{e}$ | effective shear stress | Pa | |
| $\sigma_0$ | residual stress | kPa | 10 |
| $E_\mathrm{SIA}, E_\mathrm{SSA}$ | enhancement factor for the SIA and SSA | | 1, 0.5 |
| $A$ | ice rate factor | $\mathrm{s^{-1}\,Pa^{-3}}$ | |
| $n$ | Glen flow law exponent | | 3 |
| $\boldsymbol{u}$ | horizontal SIA velocity | $\mathrm{m\,s^{-1}}$ | |
| $\boldsymbol{v}$ | horizontal SStA velocity | $\mathrm{m\,s^{-1}}$ | |
| $\boldsymbol{U}$ | horizontal hybrid velocity | $\mathrm{m\,s^{-1}}$ | |
| $v_0$ | regularisation speed in SStA equations | $\mathrm{m\,yr^{-1}}$ | 0.01 |
| $\boldsymbol{\tau}_\mathrm{b}$ | basal shear stress | Pa | |
| $N_\mathrm{b}$ | effective basal pressure | Pa | |
| $p, q$ | sliding law exponents | | 3, 2 |
| $\gamma$ | sub-melt-sliding parameter | K | 3 |
| $C_0$ | inverted basal sliding parameter | $\mathrm{m\,yr^{-1}\,Pa^{-1}}$ | |
| $\Delta t_\mathrm{inv}$ | time step for inversion of $C_0$ | yr | 50 |
| $H_\mathrm{inv}$ | scaling factor for inversion of $C_0$ | m | 5000 |
| $r$ | slip ratio of grounded ice | | |
| $r_\mathrm{thr}$ | default threshold value of $r$ | | 0.5 |
| $v_\mathrm{ref}$ | default reference value of $|\boldsymbol{v}|$ | $\mathrm{m\,yr^{-1}}$ | 100 |
| $w$ | weighting function in hybrid schemes | | |
| $h_\mathrm{rlx}$ | scaling factor for relaxation procedure | | 0.001 |





**Table 2.** Summary of the results at the end of the calibration procedure, including: Total grounded ice volume $V_{grd}$ (km$^3$); deviation from total grounded ice volume $\Delta V_{grd}$ (%); mean absolute error in the ice thickness $\bar{\Delta} H$ (m); Fraction of the total area where basal sliding is activated $A_{sld}$ (%); mean ice thickness error only where basal sliding operates $\bar{\Delta}_{sld} H$ (m); Mean surface velocity error $\bar{\Delta} v_s$ (m yr$^{-1}$); fraction of the grounded ice area dominated by the SIA ($w < 0.25$) $A_{SIA.25}$ (%); and fraction of the grounded ice area dominated by the SStA ($w > 0.75$) $A_{SStA.75}$ (%).

| Scheme | $V_{grd}$ [km$^3$] | $\Delta V_{grd}$ | $\bar{\Delta} H$ [m] | $A_{sld}$ | $\bar{\Delta}_{sld} H$ [m] | $\bar{\Delta} v_s$ [m yr$^{-1}$] | $A_{SIA.25}$ | $A_{SStA.75}$ |
|---|---|---|---|---|---|---|---|---|
| HS-1 | $2.58 \times 10^7$ | +1.22% | 49.9 | 53.3% | 28.0 | 51.0 | 44.3% | 30.2% |
| HS-2a | $2.60 \times 10^7$ | +1.88% | 59.8 | 60.4% | 26.9 | 42.5 | 80.2% | 3.2% |
| HS-2b | $2.54 \times 10^7$ | −0.25% | 52.0 | 65.6% | 35.9 | 32.9 | 81.0% | 8.5% |
| HS-3 | $2.58 \times 10^7$ | +1.23% | 54.3 | 63.5% | 31.4 | 46.2 | - | - |
| SoS | $2.51 \times 10^7$ | −1.55% | 73.1 | 61.2% | 65.9 | 80.5 | 100% | 0% |

**Table 3.** Summary of the results at the end of additional 100kyr runs with prescribed median of inferred basal sliding coefficients, including: Total grounded ice volume $V_{grd}$ (km$^3$); deviation from total grounded ice volume $\Delta V_{grd}$ (%); mean absolute error in the ice thickness $\bar{\Delta} H$ (m); and mean ice thickness error only where basal sliding operates $\bar{\Delta}_{sld} H$ (m).

| Scheme | $V_{grd}$ [km$^3$] | $\Delta V_{grd}$ | $\bar{\Delta} H$ [m] | $\bar{\Delta}_{sld} H$ [m] |
|---|---|---|---|---|
| HS-1 | $2.58 \times 10^7$ | +1.08% | 64.0 | 50.6 |
| HS-2a | $2.72 \times 10^7$ | +6.65% | 150.0 | 119.0 |
| HS-2b | $2.39 \times 10^7$ | −6.39% | 172.0 | 120.7 |
| HS-3 | $2.51 \times 10^7$ | −1.57% | 93.3 | 83.6 |





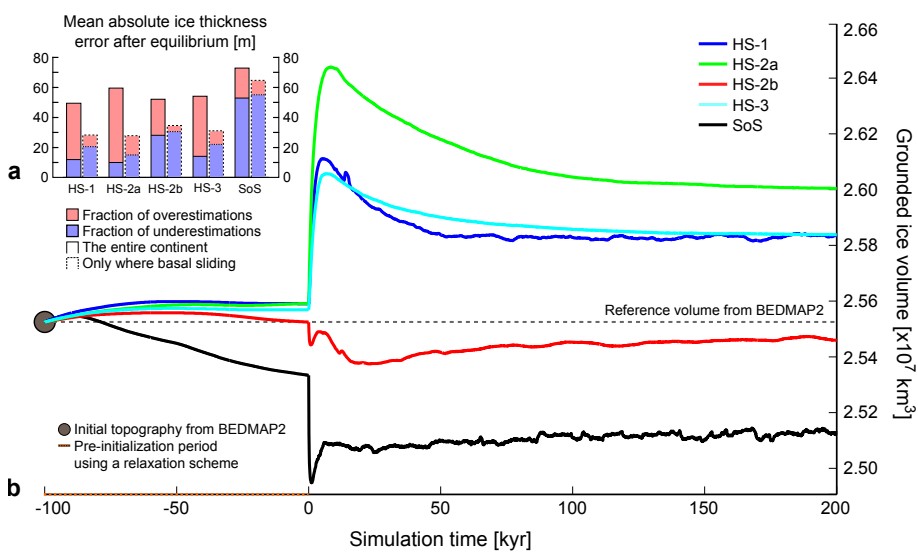

**Figure 1.** Overview of the calibration procedure. **(a)** Mean differences between the modelled and observed ice thickness at the end of the simulations, in m. For each scheme, a mean error is calculated for the entire ice sheet (left bars) and separately over the areas where basal sliding is identified (right bars). Fractions of the mean error arising from under- and overestimations are shown in blue and red, respectively. **(b)** The evolution of the total grounded ice volume during the calibration procedure, in $\times 10^7 \, \text{km}^3$.



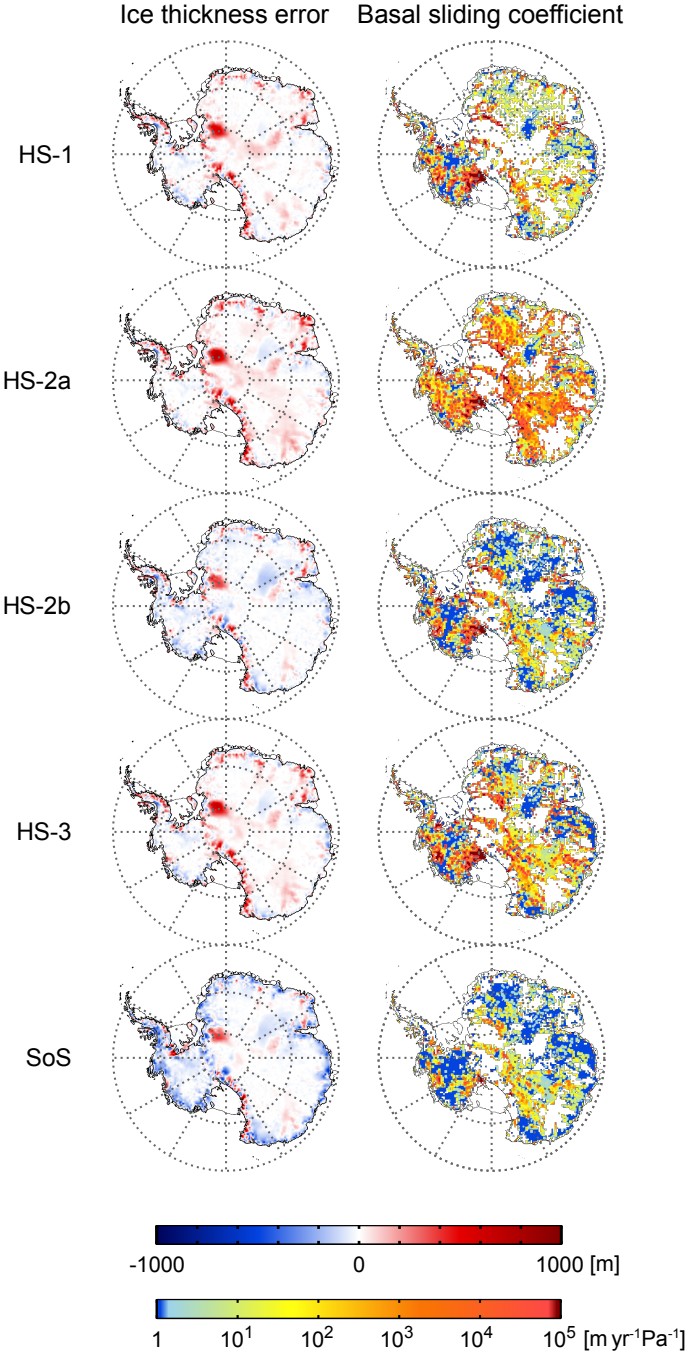

**Figure 2.** Comparison of the equilibrium ice sheet states derived from different schemes. **Left column:** Differences between the modelled and observed ice thickness, in m. **Right column:** Inferred distributions of basal sliding coefficients, in $\mathrm{m\,yr^{-1}\,Pa^{-1}}$. Non-coloured areas mark the locations where basal sliding is not identified and the calibration procedure does not operate. Color-code saturates at the upper and lower limits allowed for the inversion procedure.




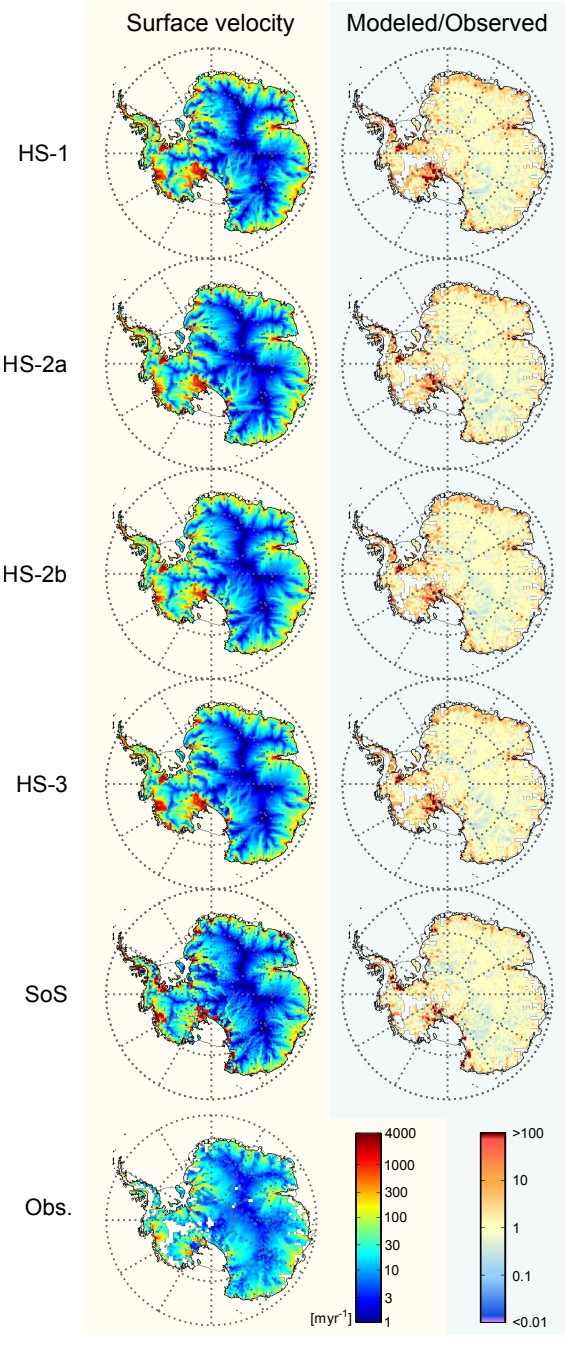

**Figure 3. Left column:** Equilibrium surface velocities across the grounded ice areas derived from different schemes, in $\mathrm{m\,yr^{-1}}$, compared to the observational data set from Rignot et al. (2011) (bottom), regridded to the model resolution of $40\,\mathrm{km}$. **Right column:** Ratios of the modelled to observed surface velocities, plotted on a logarithmic scale. Velocities smaller than $2\,\mathrm{m\,yr^{-1}}$ are excluded. Color-code saturates at ratios larger than 100 or smaller than 0.01.





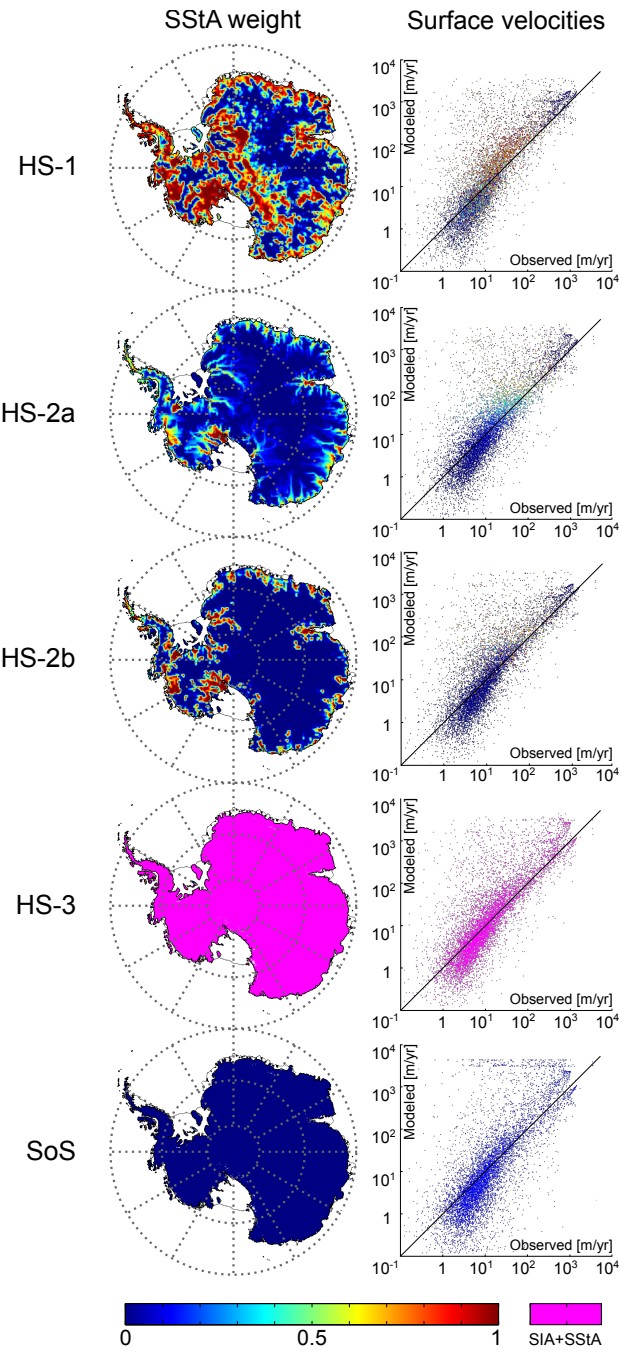

**Figure 4. Left column:** Equilibrium SStA weights derived from different schemes. The HS-3 does not use a weighting function and simply adds velocities derived from both shallow approximations. **Right column:** Scatter plots of modelled vs. observed surface velocities, in $\mathrm{m\,yr^{-1}}$. Each grid cell is color-coded according to the corresponding SStA weight.





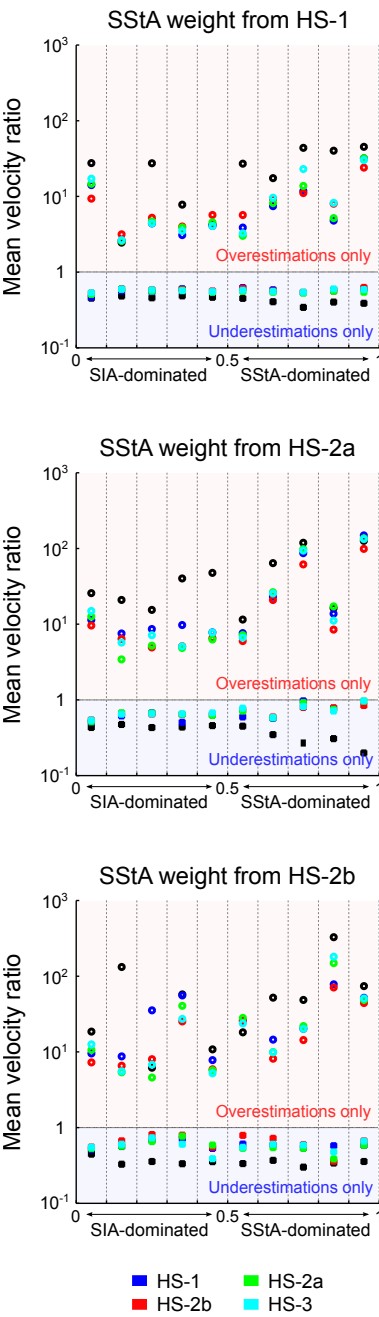

**Figure 5.** Surface velocity error as a function of the SStA weight. The errors are represented by ratios of modelled to observed velocities (Fig. 3). To enable a comparison of all schemes, the distributions of $w$ derived by HS-1, HS-2a, and HS-2b are used to quantify different ice flow regimes. Each circle spans a weight interval of $0.1$ and represents an average of velocity ratios computed using grid points with $w$ values within this interval (Fig. 4). The averaging procedure is performed separately for under- and overestimations.





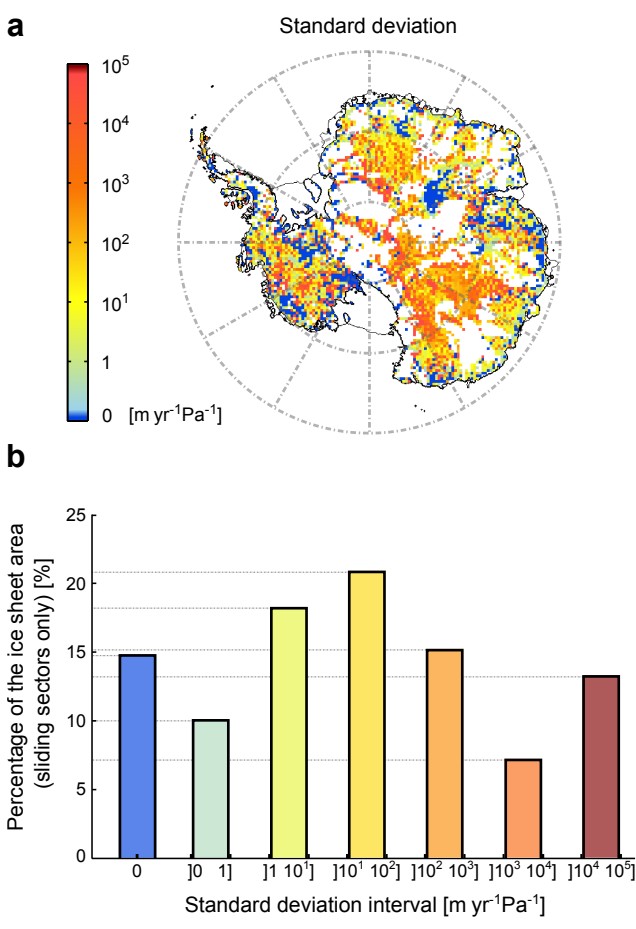

**Figure 6. (a)** Standard deviation of the retrieved distributions of basal sliding coefficients, $C_0$, in $\mathrm{m\,yr^{-1}\,Pa^{-1}}$, from four hybrid schemes. **(b)** Fraction of the total sliding area, in %, characterised by different magnitude ranges of the standard deviation of $C_0$.





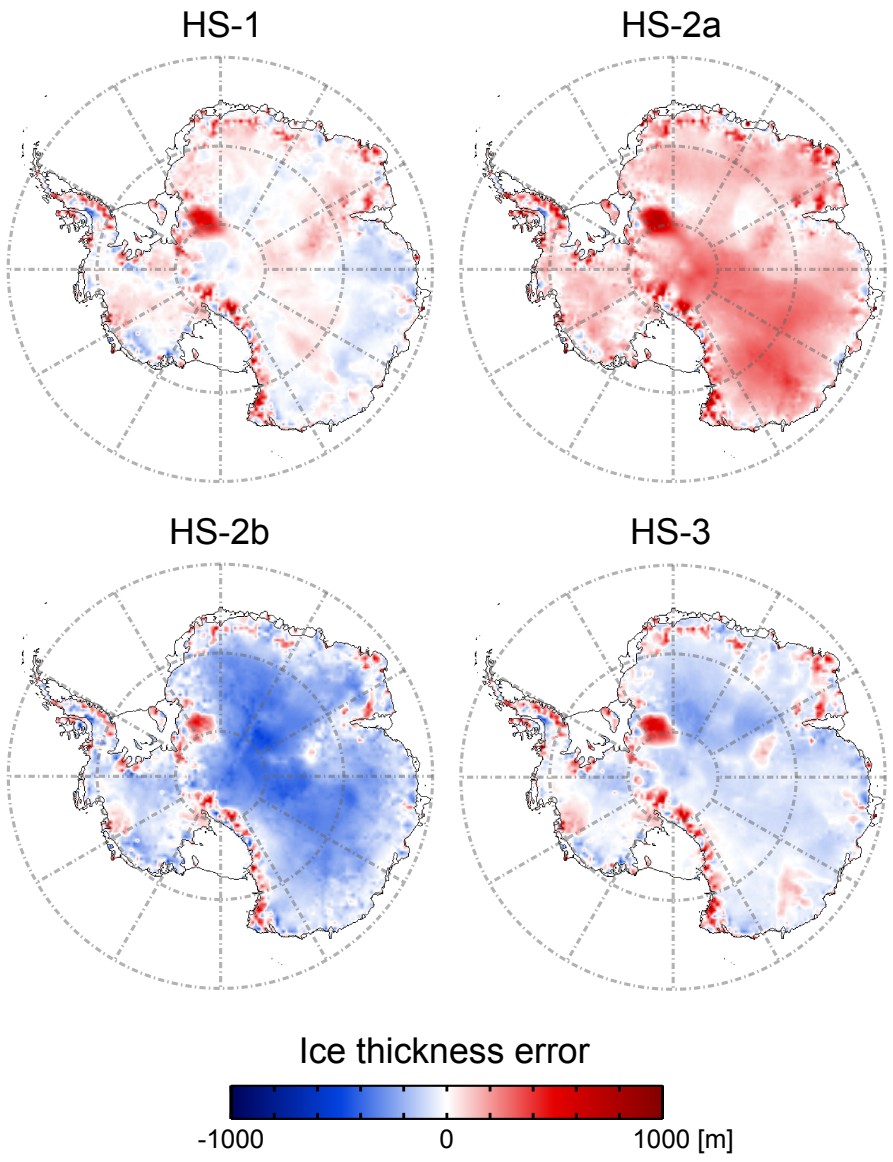

**Figure 7.** Difference between the modelled and observed ice thickness for all hybrid schemes, in $\mathrm{m}$, after an additional $100\,\mathrm{kyr}$ run in which a median of the inferred distributions of $C_0$ is prescribed at the ice sheet base. The values of $C_0$ derived from the SoS are not included in the median distribution.





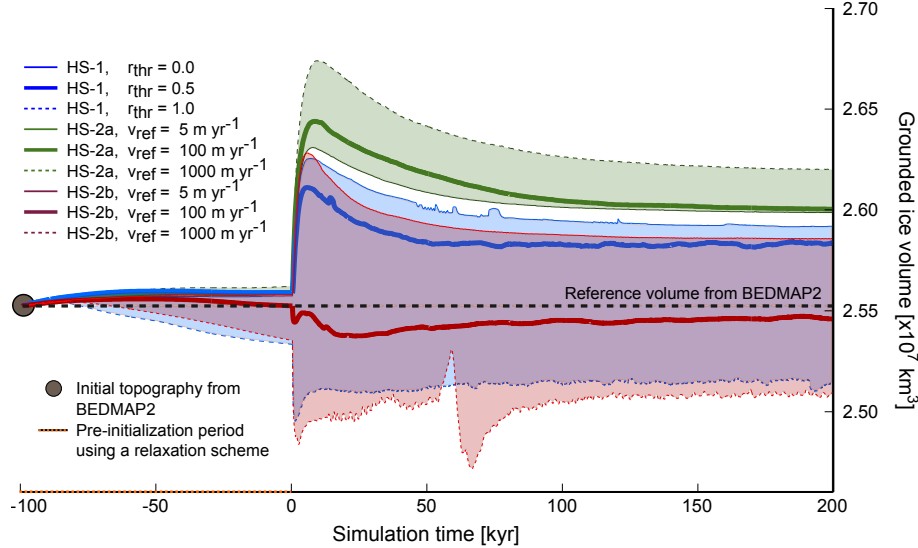

**Figure 8.** The evolution of the total grounded ice volume during the calibration procedure, in $km^3$, for different values of free parameters included in the computation of the SStA weight. The volume spread contains the reference values used for the experiments discussed in Sections 4.1–4.3 (solid thick lines), and values that are representative of the lower and upper limits of possible parameter ranges (thin solid and dashed lines, respectively).

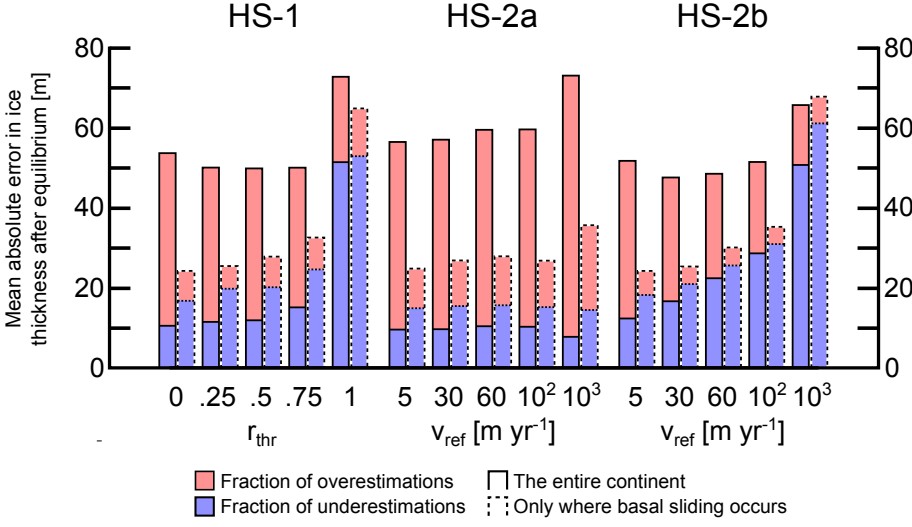

**Figure 9.** Mean differences between the modelled and observed ice thickness throughout the simulations, in $\times 10^7 \, km^3$, for different values of free parameters included in the computation of the SStA weight. As in Fig. 1, a mean error is calculated for the entire ice sheet (left bars), and separately over the areas where basal sliding is identified (right bars). Each bar is divided into fractions of the error arising from under- (blue) and overestimations (red).