# Peer review of "Comparison of hybrid schemes for the combination of Shallow Approximations in numerical simulations of the Antarctic Ice Sheet"

_The Cryosphere, 2016_

## Referee Comment (RC1) · Anonymous Referee #1 · 27 Jul 2016

General comments

This paper describes a study of 4 hybrid schemes, combining the Shallow Ice and Shelfy Ice approximations for Antarctic Ice Sheet simulations. The 4 schemes are implemented into the open source ice sheet model Sicopolis (Greve, 1997), and a number of simulations are made for the Antarctic Ice Sheet with each and the resulting basal sliding coefficients compared. Analysis of the calibrated sliding parameter is done and runs where average values from all 4 schemes is applied, as well as the results from other schemes (swapped). The paper is clear but the conclusions are not clearly specified and reader is left with wondering if authors have been able to conclude which scheme is preferred and how they will continue working with one, or all, of the

schemes. A clearer message and conclusions from the study, as well as more concise analysis of the results (e.g. Figures 5 and 6) would improve the paper.

The structure of the paper could improve by adding a separate data section, before the Methods section, the text in lines 4-22 on page 8 would be better located in a separate Data section and a more detailed information about the data used for this modelling approach would be beneficial for the paper. Also lines 19-24 on page 9 would be better located in a data section.

The wording of the method used and the results is confusing, in this study a forward method is used to determine or rather calibrate Co , the basal sliding coefficient, and therefore it is confusing to call it inversion technique (e.g. line 23 page 8) or inversion (e.g. line 28, page 8), suggest to call it iterative technique or calibration, see further comments below (as in line 5 in the abstract: "the model is calibrated using an iterative technique . . .")

It is not clearly explained why authors are swapping the determined Co values between the calibrated models, why is this useful? Is the following ice sheet adjustment indicating model consistency? As stated in the paper, each method has its own way of combining the approximated term and not clear why swapping of the determined (or calibrated) Co values would give any insight in the model behavior or result. It is discussed that the different methods, and the results of Pollard and DeConto (2012) give qualitative similar results, with high/low values in same regions, but the numerical value of Co is method/model dependent and not clear what swapping of the results is useful for. The text on page 15 is not clear and the two figures 5 and 6 are not discussed in satisfying manner, what information about the schemes and the results can we draw from this analysis?

Specific comments:

The wording in abstract and introduction is confusing, the Shallow Ice approximation is a zeroth order approximation assuming the thickness of the ice is much smaller than

the length scale and thereby horizontal stress gradients omitted. This approximation has no assumptions for the sliding law and a full system model would equally have to assume some sliding approximation, or shelfy ice solution to account for sliding. Line 2 in the abstract should be reworded (SIA is not applicable . . . where basal sliding operates) and line 6 (minimal sliding) on page 2. Consider to rewrite also lines 2-5 on page 2, ("neglecting terms" and "simplest and most commonly used"): the Shallow Ice Approximation is a zeroth order approximation of the momentum balance equation assuming the H/L is very small.

The wording in the abstract in lines 9 and 11 is not clear and should be rewritten for clarity: "averaged and swapped" cannot be understood until reading the main text of the paper and therefore needs some clarification in the abstract. ".. this requirement for internal consistency" – is not clear in this context and needs more explanation.

I find missing an overview figure indicating the location of the areas that are named in the paper, such as Dronning Maud Land, Coats Land, Siple Coast etc., this would be useful for readers not familiar with place names in Antarctica and makes the paper easier to read.

Note that "Blatter-Pattyn models" were developed much before the asymptotic analysis by Schoof and Hindmarsh (2010) and therefore it would be appropriate to reference the earlier papers with these model developments.

Lines 27-29 on page 2 are not clear, more information and detailed explanation of what authors mean is needed here, why would it be necessary to look for further explanations when two models yield similar result under similar forcing?

Model resolution is very low, were any tests made to assess the sensitivity of the method to grid resolution? - or to potential errors/inaccuracies in the topographic data? How good is the observed topography?

The quality of English is generally good, but in many places the wording is strange and

needs some editing, it would be beneficial for the paper to have a thorough editing of all the text. Below are a number of places indicated where rewriting/editing would improve the quality of the text.

Technical corrections:

Page 1, line 2, suggest to replace "ice dynamics" with "stress within the ice" Page 2, line 10, suggest to add "at the base" after "no friction" Page 2, line 21, suggest to rewrite, models do not "detect", replace with something like "the algorithm used to identify . . ." Page 2, line 23, suggest to replace "versus" with "compared to" Page 2, line 25, suggest to replace "superposition" with "combination" Page 2, line 31, suggest to rewrite, replace "these result from .." with " these are . . ." Page 2, line 32-33 reword sentence: "Mechanical properties . . . . may serve as an example of . . . parameters" does not make sense. Page 2, line 34, something missing "widespread misfit" of what? Elevation? - do all models show this type of misfit? A reference to a study showing this would be useful here. Page 3 Line 11, missing information and reference, what observational data sets are used? Line 14, add "the" between by and different Line 19, see comment above, replace "inverse" with "iterative" Line 29, the model does not "consider" or is "keeping track" of temperature, the algorithm or post processing does, suggest to rewrite. Suggest to add "computed" after "ice is"

Page 5 Line 1, see comment above, "detect" seems a strange selection of word, suggest to replace with "determine" or "identify" Line 14, not clear wording "a consistent use of inverted distributions of Co" - (replace inverted with determined, or calibrated) Line 21, not clear wording: "enters the computation of SStA velocities" – suggest something like "and SStA velocities are computed for this point" Line 25, suggest to delete "which is assigned to the SStA" this is explained in next sentence (and add velocity after SStA)

Page 6, Line 2, suggest to add "SIA and SStA" before "Velocities are . . ." Line 16-17, unclear wording, suggest something like "It is rather used to determine. . . . . . the computed SStA contribution should partly or completely replace sliding" Line 24, sentence is not clear, suggest to rewrite, something like "In the continental interior the modelled ice flow is dominated by the SIA solution Line 29 see comment above, replace "inversion" with "determination or calibration" Line 30, replace "infer" with "determine"

Page 7, Line 11, see comment above, "easier activation of the inversion procedure" suggest to rewrite to something like "more frequent computation of sliding velocity" - what does "slightly" mean here? Line 14, suggest to delete "similarly, and" Line 16, suggest to replace "speed" with "velocity", not clear what "local adjustment" means here, elevation, or Co? suggest to replace "keeps the inversion" with "prevents the method" Line 17 suggest to replace "over-adjustment of" with "over-adjusting", replace "speed" with "velocity" Line 18, suggest to delete "the" before "numerical" Line 20, replace "This" with "These" (plural of values) Section in lines 22-29 is not clear, and needs rewriting for clarification what is the time in the iterative method or calibration? Line 24, "time derivative of the ice thickness", do you mean observed or modelled? why suspend adjustment if previous time step reduced the difference? (line 24): What process? Line 26 replace "overshoot prevention" with "over-adjustment of Co" line 27, suggest to replace "lets" with "allows" and delete "to" before "influence" line 31, add "the" before "fringing"

Page 8 See comment above, move lines 6-18 to a Data description section Line 6, replace "which" with "that" Line 19 suggest to add "is used to" before "account", replace "changes" with "discrepancies" , replace "by" with "with" Line 23, see comment above, replace "inversion" with "iterative" or calibration Line 24, this is forward method, suggest to replace "inversion" with "iterative" Line 28, replace "inversion" with "calibration" see above Line 30, suggest to delete "one-to-one" Line 33, suggest to replace "not accounted for" with "is not included in the simulations"

Page 9 Line 1, see above, replace "inversion" with "calibration" Line 12, replace "simulations" with "calibration run" Line 13, see above, replace "inverse technique" with "iterative" or calibration Line 27 what do you mean by "glacier flux gate" do you mean

an outlet glacier? Line 21 not clear text "which overlap at their interface" needs clarification here

Page 10 Line 8, suggest to add "a" before "new" Line 12-13, this sentence is not clear and needs rewriting (what is internal operation of the hybrid schemes?) Line 17, what does "quasi-equilibrium" mean here? Line 19, what is "negligible" in this context? A percentage or some value would be useful here Line 24, suggest to replace "prevents" with "does not require"

Page 11 Line 9, the smallest error of 49.9m is according to the table using HS-1, is there an error in the table? If not then this section should be rewritten to reflect that. Line 21, replace "independently" with "independent" - here some discussion would be appropriate about if this is related to the common SMB forcing or geothermal heat flux? Line 22, add "simulated" before "frozen" Line 24, suggest to edit, change to something like: "far below the pressure melting point" and delete "the" before "white coloured" Line 25, edit the text, delete "on the other hand", suggest to write "Areas where ice is underestimated . . ." – but what does "sparsely distributed" mean? Line 27 replace "inversion" with "determination" or "computation" Line 31, replace "inversion" with "iterative" or calibration

Page 12 Line 2, replace "inverted" with "applied" or "computed" Line 4-5, same comment, suggest to write "region where Co is not applied" Line 6, replace "inversion" with "calibration" Line 21, delete "also" Line 26, what do you mean here? What significant modification of Pollard and DeConto (2012) scheme would result in similar values of Co? suggest to replace "perturbation" with "distribution" or "pattern" and delete "inverted" Line 30 add "the" before hybrid Line 33 replace "small" with "low" and add "the" after "near" Line 35, what are "high velocity flanks"? suggest to add "sheet" before "margins" and at the end of line, after "ice"

Page 13 Line 2, strange wording, suggest to replace "contaminated by" with "characterized with" Line 6, add "the" before hybrid Line 7, suggest to replace "in" with "at" Line 8,

"flux gates" - not clear, is this a specific location? Line 11, delete "inferred" and delete "On the other hand" Line 14, delete "inferred" Line 19, delete "a" before slow ice motion, replace "flow speed" with "velocity" and "predicted" by "simulated" Line 21, delete "Arguably" Line 22, replace "stagnated" with "stagnant" Line 25, not clear wording "deviated to either side" - "pushed to merge" are you referring to modelled or observed velocity? – what side of what?, what is pushing what?, suggest to replace "deficiencies" with "errors" Line 28, according to the table, it is HS-1 that has the minimum misfit for the ice thickness Line 30, see above, replace "inversion" with "calibration" Line 31, suggest to delete "in the modelled surface velocities" Line 33, what do you mean by "opposite ends"? Line 34, suggest to replace "enabling" with "adding"

Page 14 Line 4, suggest to replace "imply" with "control" Line 5, what does "internal operation of the hybrid scheme" mean? Line 1, suggest to replace "the velocity field from the observational data set" with " the observed velocity" Line 12, suggest to replace "simple differentiation" with "transition" Line 17, reword, "tends to prevent" to "can prevent" and delete "s" in causes (and cause underestimation..) Line 24, what is a "cursory comparison" ? clarification is needed Line 29-30 suggest to rewrite (delete Here we attempt) and write: To isolate the influence . . .. we plot averaged errors . . . Line 35, replace "inferred" with "resulting" or delete "inferred"

Page 15 Line 1 add "the" before different Lines 2-5 this sentence is not clear and does not explain the difference between the 3 figures in Fig.5 clarification is needed here Line 9, suggest to replace "looking" with "comparing" Line 19 delete "the" before "period" Line 26 – this line is not clear, what do you mean by "generalizing" – see comment above, it is not clear what information about the model can be drawn from Figures 5 and 6 Line 34, delete "On the other hand,"

Page 16 Line 2 suggest to add "distributions of Co from" before the HS-1 and HS-3 Lines 1-6, it is not clear from this text what this analysis gives for useful information about the different schemes used in the study Line 8, add "The" at beginning of line Line 11, rewrite: "allows us" is a strange wording here Line 23, what is the criteria

for selecting lower and upper limits for each scheme? Line 31, suggest to replace "decrease" with "get worse"

Page 17 Line 4, delete "by" before 2.5% Line 24, delete "," after "enables" Line 27-29 suggest to rewrite, something like "differ in the way ways the 1) relative contributions . . . are computed 2) areas where . . . is applied is determined, and 3) basal sliding is accounted for. Line 30, see above, replace "inverse" with "iterative" or "calibration" Line 31, delete "all" or add "the applied" after "all" Line 32, add "the" before "schemes", replace "below" with "less than" and add "the" before "total"

Page 18 Line 3, delete one "can" Line 5, add "the hybrid" after "all" Line 7, add "the" before hybrid Line 8, suggest to replace "develops" with "exists" Line 15 What particular scheme is discussed here? It is not clear Line 19 add "," after Here Line 27, what does "the neglect of paleoclimate signal" mean here – can applied geothermal heat flux, or applied SMB play a role here? Line 33, does the scheme affect the way the temperature is computed? Line 34, does the scheme allow determination of hard or soft bed? – it can only indicate high or low value for Co, or what?

Page 19 Line 3, suggest to replace "inferred" with "calibrated" - not clear how high variability can "provide an opportunity to quantify the effects of the uncertainty" – suggest to rewrite to clarify what is meant here. Line 8, see comments above, replace "inverse method" with "iterative technique" Line 9, what is meant with "internal consistency required to avoid misfit" suggest to rewrite to clarify. Suggest that the concluding sentence of the paper will state the main results of the study and how it can be useful for further modelling approaches.

Figures and tables

Table 2, suggest that the caption include some explanation, referring to text, what the different lines stand for (HS-1 etc). See comments above, the lines may have got mixed up, since the text states that the minimum difference for elevation is for HS-2, but 49.9 m is in line HS-1

Figure 2, is this at the end of the simulation?

Figure 4, suggest to replace "grid cell" with "point" for the right hand column figures

Figure 5, the figure caption is not clear and needs editing. What does "quantify different ice flow regimes" mean here? The y-axis label is mean velocity ratio, but the text states Surface velocity error, clarify what is shown here. Suggest to replace "by" with "as" before ratios.

Figure 6, as discussed above it is not clear how this figure is useful. What information can be gained from this analysis? "retrieved distributions of basal sliding coefficients" is not clear, do you mean standard deviation of the determined values with each scheme in each grid point?

Figure 7, what does this figure tell us? What meaning does the median of the inferred distributions of Co have? Is this a useful quantity?

Figure 8, what do the bumps in the lines of HS-1 with r-thr=0.0 and HS-2b for v-ref=1000 m/yr shortly after 50 kyr mean? Is this instability in the simulations?

Figure 9 "throughout the simulations" – do you mean at the end of simulations?

Please also note the supplement to this comment:
http://www.the-cryosphere-discuss.net/tc-2016-117/tc-2016-117-RC1-supplement.pdf

―――――――――――――――――

---

## Referee Comment (RC2) · Anonymous Referee #2 · 18 Aug 2016

The manuscript by Bernales et al uses an ice-sheet model for Antarctica to test different schemes for the calculating of the ice velocities. The main focus is on combining the shallow ice and shelfy stream approximations and use different hybrid schemes to combine the two. Four different schemes are implemented in the model and used to inversely determine the basal sliding coefficient C0 by looking at the difference between modelled and observed ice thickness. Differences between the four schemes are explained clearly, through how the spatially varying basal sliding coefficient differ for each scheme, how grounded ice volume evolves and how ice thicknesses vary relative to the observations. Model performance is also checked against observations of ice surface velocities. Methods and results are explained clearly. The two main concerns I have

are that the manuscript does miss a real strong final conclusion and that uncertainties in the methodology should be discussed in more detail.

**Main comments**

1. Uncertainties in methodology

Paragraph starting on Page 2, Line 30 to Page 3, Line 8:

You should here also mention that there are more parameters and/or processes involved that are poorly constrained (or less known) such as bedrock uplift due to postglacial rebound (GIA), response of the bed itself to current changes in ice, heat flux from the bed, properties of ice flow (flow parameter) or the bedrock height uncertainty (see Bedmap2 paper). Following on the uncertainties, an important point to make using this method is that by solving for the basal stress you actually compensate/correct for other (some of them mentioned above) uncertainties in the boundary conditions and uncertainties/errors in the model itself. This is I think a key point to keep in mind when using the methods described in the manuscript. This should be mentioned here and discussed thoroughly in the final discussion/conclusion as well!

**2.** Rewording of basal velocities and SIA Starting on Page 4, Line 16:

Throughout the manuscript it is mentioned that SIA has a basal velocity. This is mostly a matter of rewording some of the text, not so much an error but rather how this is actually computed. I think the SIA itself does not have a basal velocity, but rather that the basal velocity is a basal boundary condition to calculating the SIA, which could also be set to zero if you do not include sliding at all. This is actually nicely illustrated in Fig. 1 of Bueler and Brown (JGR, 2009). I think this could be rewritten in some parts of the manuscript. In the general comments below some of these lines are mentioned.

3. Final conclusions On page 19:
The sentence "This suggests .. Stokes equations." seems quite obvious but good to mention. However in the final conclusions I do miss a discussion on two things particularly:

- Is a distribution of basal parameter C0 derived using present-day forcing and observations also applicable for long-term paleoclimate simulations, can it be used for other time periods and climate regimes?

- What is the best scheme to be used here? Is there a scheme that simulates the observations best, also considering the lowest over- and underestimations, i.e. the lowest absolute difference in ice thickness.

**General comments**

Line 1: Replace introduced with used or implemented

L 5: What do you mean with realistic scenario? Do you mean like present day climate forcing? Please be specific.

L 8: Remove comma after Despite this

L 8: Robust agreement with what, or of which variable? Again please be specific.

L 17: Change to: However, the time scales over which an ice sheet builds up and disintegrates

L 9: Remove commas before membrane and after important

L 7-14: You could first introduce the SSA (L10-14) and then discuss the ice streams (L7-10).

L 15: Instead of mentioning highly dynamic regions, actually state here something like: The limitations of SIA models for calculating the highly dynamics ice streams have ..

TCD
L30: Rephrase realistic scenarios, same as in the abstract, present-day climate forcing?

L 17: No new paragraph, start sentence as: First, the ice-sheet model

**Page 4**

L 16: Please rewrite, the SIA itself does not have a basal velocity, but rather U-b is here the basal boundary condition to calculating the SIA.

L 18: Eexplain here how the effective basal pressure Nb is calculated.

L 22: Is Tm an actual temperature or a temperature difference. Explain this in the text.

**Page 5**

L 19: Rewrite to something like: where ub is the Weertman sliding velocity (Eq. (2)) and us the surface SIA velocity, respectively.

**Page 6**

L 11-16: Do not mention sliding-SIA but rather SIA including (Weertman) sliding. Here it could also be discussed the carefulness of using Weertman sliding, as discussed by Bueler and Brow (JGR, 2009; see also their Appendix B) it leads to a discontinuities in the velocity field.

L 16: change to: where ub is the basal sliding velocity as in ..

L 28: At the end of section 2.2 a table could be added that summarises the 4 schemes with the column show something like : 1) name, 2) sliding 3) reference to equations, 4) reference to studies

**Page 7**

L 23-24: Can you explain here and perhaps clarify in the text why you stop the adjustment when the difference between modelled and observed ice thickness is reduced?? A reduction of this difference is actually what you want right?
**Printer-friendly version**

L 22-29: This paragraph could be replaced to Section 3.

**Page 8**

L 13: On spin up procedures for ice-sheet models see also this paper, you might want to refer to this.

Fyke, J. G. and Sacks, W. J. and Lipscomb, W. H., A technique for generating consistent ice sheet initial conditions for coupled ice sheet/climate models. Geoscientific Model Development, 7, 1183 - 1195, 2014.

L 15-16: Reword: a simulation time of 100.000 years (100 kyr)

L 21-22: Spanning 100 kyr (mention exact length of your experiments)

**Page 10**

L 1: Remove -and get insight into-

L 4: Be specific: observed ice-sheet thickness and surface velocities as a measure ...

L 19: .. the variations of the total grounded ..

L 26: Numerically more or equally stable, and also a similar computational time compared to the SoS?

L 1: Change to: that include basal sliding with the SIA solution

L 4: Remove -high frequency-

L 7: The inset of Figure 1

L 13: Change to: that include basal sliding with the SIA.

L 20-21: The large overestimation of ice thickness between Shakleton Range and the Pensacola Mountains as you state is seen in all panels of Fig. 2. Could you give a more clear explanation why this particular area is overestimated. Also note that this is
a region of large uncertainties in ice thickness (see Bedmap2 paper).

L 28, 32: I suggest not to refer to Fig. 1a but to: the inset of Fig. 1.

L 2: Change to: over the areas where there is no sliding and C0 is not inverted.

L 32: Change to: .. that are unresolved by the model due to its..

L 34: Remove comma after divides.

L 7: Remove commas before and after for example

L 26: Change to: .. projection onto the course horizontal grid we use here.

L 29: Remove comma after simulation

L 33: Change to: Equation (10) uses the basal velocities from equation (2) to compute

••

L 34-35: Where sliding velocities from equation (2) are high.

L 1: Here also perhaps mention how H-2a compares to H-2b?

L 5: are implemented

- L 5: Change to: .. hybrid schemes, Fig. 4 illustrates the
- L 22: Remove comma after general
- L 23-24: At a first glance it may
- L 25: Change to full stop: the case.

L 2: Change to: The misfit for Hs-1 increases ..
L 4: The Hs-3 experiment shows an ..

L 19: Change to: the basal velocity in the Hs-2b experiments

L 24: Remove commas before and after continental-scale.

L 25: Remove entire

L 1: Also mention the range of the ice thickness errors not only the means.

L 17: Change to: which adds up SIA (without sliding) and SstA.

L 29: Also mention the uncertainties in ice thickness and surface velocities (see their respective references).

**Tables and Figures**

Table 2:

Should the sum of the last two columns not be 100%? Please clarify and explain.

Fig. 1:

- Label a and b can be removed, rather refer to fig. 1 and inset of fig. 1.

- Instead of a coloured bar for the pre-initialization the names of the periods could also be placed above (pre-initialization period and automatic calibration).

- In the legend of the inset the difference between solid and dashed should be clearer.

Fig. 4:

It is not really clear which scheme represents the surface velocities the best. Could you also add a correlation/R-square value to the scatter plots? Also use the correlation in the discussion.

Fig 8:
Hs-2a always seems to be have a misfit, whereas the other scheme should have a possible solution which closely fits the observations, for a particular (probably not the same) set of parameters. Please explain and discuss this in the conclusions.

TCD

---

## Author Comment (AC1) · 4 Oct 2016

Responses to review 1 of the manuscript "Comparison of hybrid schemes for the combination of Shallow Approximations in numerical simulations of the Antarctic Ice Sheet ":

General comments

This paper describes a study of 4 hybrid schemes, combining the Shallow Ice and Shelfy Ice approximations for Antarctic Ice Sheet simulations. The 4 schemes are implemented into the open source ice sheet model Sicopolis (Greve, 1997), and a number of simulations are made for the Antarctic Ice Sheet with each and the resulting basal sliding coefficients compared. Analysis of the calibrated sliding parameter is done and runs where average values from all 4 schemes is applied, as well as the results from other schemes (swapped).

We thank the reviewer for their very constructive comments that helped to improve the clarity of the manuscript overall. Our replies are provided in blue.

The paper is clear but the conclusions are not clearly specified and reader is left with wondering if authors have been able to conclude which scheme is preferred and how they will continue working with one, or all, of the schemes. A clearer message and conclusions from the study, as well as more concise analysis of the results (e.g. Figures 5 and 6) would improve the paper.

We agree. We have modified the text to include our conclusions regarding the relative performance and the potential for future applicability of each scheme. Our evaluation criteria do not only include the fit of the results to observations, but also the numerical stability of the schemes. Our experiments at higher grid resolutions show that the HS-1 and HS-2b schemes become numerically unstable, due to large gradients in basal sliding coefficients arising from the use of basal velocities as boundary conditions for the SIA solutions in conjunction with the calibration of sliding coefficients. The HS-2a and HS-3, which utilize the SStA as a sliding law, are numerically more stable to variations in model parameters and changes in grid resolution (especially HS-2a, which rarely produces simulation crashes). The drawback of the HS-2a is in its limited ability to influence the fit between the modeled and observed ice thickness in ice sheet sectors where the SStA velocities are low (<100 m/yr), which is the case over large tracts of the Antarctic interior. The HS-3 overcomes this limitation by accounting for the SStA contribution everywhere, but so it does with the SIA velocities, which in certain areas such as, e.g., the steep ice sheet margins are excessively high, as shown in Figure 3 of the original version of the manuscript for the SoS. To improve the performance of both schemes, our future work will reconcile the drawbacks of HS-2a with the advantages of HS-3, providing a very stable and flexible hybrid scheme. Following the suggestions of the reviewer, we have tried to make the results section more concise (see the related point below).

The structure of the paper could improve by adding a separate data section, before the

Methods section, the text in lines 4-22 on page 8 would be better located in a separate Data section and a more detailed information about the data used for this modeling approach would be beneficial for the paper. Also lines 19-24 on page 9 would be better located in a data section.

Done as suggested. The data sets are now described in a greater detail in a separate section, including their respective uncertainties. To keep the flow of the text, we have located this new section after the description of the model and before the description of the experimental setup.

The wording of the method used and the results is confusing, in this study a forward method is used to determine or rather calibrate Co , the basal sliding coefficient, and therefore it is confusing to call it inversion technique (e.g. line 23 page 8) or inversion (e.g. line 28, page 8), suggest to call it iterative technique or calibration, see further comments below (as in line 5 in the abstract: "the model is calibrated using an iterative technique . . .")

Done. All instances have been replaced as suggested.

It is not clearly explained why authors are swapping the determined Co values be- tween the calibrated models, why is this useful? Is the following ice sheet adjustment indicating model consistency? As stated in the paper, each method has its own way of combining the approximated term and not clear why swapping of the determined (or calibrated) Co values would give any insight in the model behavior or result. It is dis- cussed that the different methods, and the results of Pollard and DeConto (2012) give qualitative similar results, with high/low values in same regions, but the numerical value of Co is method/model dependent and not clear what swapping of the results is useful for.

These experiments are performed to show the effects of prescribing a distribution of sliding coefficients derived from one ice model (with a specific scheme/approximation for the stresses) into a different ice model. Thus, these results are indicative of the degree of inter-model consistency. This concerns not only differences between the basal sliding approaches implemented in each model, but also models using the same sliding law as part of different hybrid schemes. Given an increasing number of studies attempting to quantify the basal conditions under ice sheets through a variety of methods including ice flow models, our experiments show that one need to be careful when using these results as input data sets in glaciological models. The text of the Results and Discussion section has been modified to clarify this.

The text on page 15 is not clear and the two figures 5 and 6 are not discussed in satisfying manner, what information about the schemes and the results can we draw from this analysis?

The text has been modified to clarify this (see previous point). To provide a clearer and more concise analysis of our results, we have removed figures 5 and

6, since they do not provide the information that is not already included in the main text.

Specific comments:

The wording in abstract and introduction is confusing, the Shallow Ice approximation is a zeroth order approximation assuming the thickness of the ice is much smaller than  the length scale and thereby horizontal stress gradients omitted. This approximation has no assumptions for the sliding law and a full system model would equally have to assume some sliding approximation, or shelfy ice solution to account for sliding. Line 2 in the abstract should be reworded (SIA is not applicable . . . where basal sliding operates) and line 6 (minimal sliding) on page 2. Consider to rewrite also lines 2-5 on page 2, ("neglecting terms" and "simplest and most commonly used"): the Shallow Ice Approximation is a zeroth order approximation of the momentum balance equation assuming the H/L is very small.

We have reworded all instances in the text as suggested.

The wording in the abstract in lines 9 and 11 is not clear and should be rewritten for clarity: "averaged and swapped" cannot be understood until reading the main text of the paper and therefore needs some clarification in the abstract. ".. this requirement for internal consistency" – is not clear in this context and needs more explanation.

Done as suggested. The text as been modified to clarify this.

I find missing an overview figure indicating the location of the areas that are named in the paper, such as Dronning Maud Land, Coats Land, Siple Coast etc., this would be useful for readers not familiar with place names in Antarctica and makes the paper easier to read.

The location map has been added as suggested.

Note that "Blatter-Pattyn models" were developed much before the asymptotic analysis by Schoof and Hindmarsh (2010) and therefore it would be appropriate to reference the earlier papers with these model developments.

Done.

 Lines 27-29 on page 2 are not clear, more information and detailed explanation of what authors mean is needed here, why would it be necessary to look for further explanations when two models yield similar result under similar forcing?

We mean that if the ice models are different (e.g. as in the hybrid schemes presented in this study), each model can independently calibrate its parameters to arrive at a similar solution (as shown in our results). Since the idea provided

by this line is discussed later in the main text, we have removed it from the introduction to avoid confusion.

Model resolution is very low, were any tests made to assess the sensitivity of the method to grid resolution?

Following the concerns of the reviewer, we have managed to rerun all experiments at a higher resolution of 20km. Additionally, we have performed some sensitivity tests at higher resolutions of 15 and 10km, which show that the modeled ice flow near the ice sheet margins benefits from a denser grid due to the added flexibility (i.e. more grid points) provided to the calibration procedure, further (but slightly compared to the change from 40 to 20km) decreasing the misfit between the modelled and observed fields, representing a quantitative improvement that does not impact our conclusions.

- or to potential errors/inaccuracies in the topographic data? How good is the observed topography?

The sensitivity of the method to various model and data uncertainties (including grid resolution) was assessed by Pollard and DeConto (2012). For the topographic data, they perturbed the bedrock elevations using Gaussian noise with different noise amplitudes, finding that widespread errors of ~400m or more are necessary to considerably affect their results. In BEDMAP2, this level of uncertainty is found primarily in two regions: Between the Recovery and Support Force glaciers, and in  Princess Elizabeth land (Fretwell et al., 2013), which are characterized by unrealistically smooth topography. In the former region, we attribute to this smoothness the overestimation of ice thickness found in all the applied hybrid schemes, which are not able to reproduce the observed ice streams without the proper topographic routing.

The quality of English is generally good, but in many places the wording is strange and needs some editing, it would be beneficial for the paper to have a thorough editing of all the text. Below are a number of places indicated where rewriting/editing would improve the quality of the text.

The revised manuscript will undergo the suggested check. We have applied all the corrections suggested by the reviewer.

Technical corrections:

Page 1, line 2, suggest to replace "ice dynamics" with "stress within the ice"

Done.

Page 2, line 10, suggest to add "at the base" after "no friction"

Done.

Page 2, line 21, suggest to rewrite, models do not "detect", replace with something like "the algorithm
used to identify . . .."

Done.

Page 2, line 23, suggest to replace "versus" with "compared to"'

Done.

Page 2, line 25, suggest to replace "superposition" with "combination"

Done.

Page 2, line 31, suggest to rewrite, replace "these result from .." with " these are . . .."

Done.

Page 2, line 32-33 reword sentence: "Mechanical properties . . .. may serve as an example of . . . parameters" does not make sense.

Done.

*These limitations include the scarcity of observational data needed to reduce the errors introduced by poorly constrained model parameters, e.g., the distribution of water-saturated sediments at the ice sheet base and their potential to enhance basal sliding.*

Page 2, line 34, something missing "widespread misfit" of what? Elevation? - do all models show this type of misfit? A reference to a study showing this would be useful here.

The text has been modified to clarify this:

*This is currently considered to be a major source of large, widespread misfits between the observed and modelled elevations of the AIS (e.g., de Boer et al., 2015) .*

Page 3 Line 11, missing information and reference, what observational data sets are used?

The text has been modified to clarify this:

*[...] using state-of-the-art observational data sets of ice thickness (Fretwell et al., 2013) and ice surface velocities (Rignot et al., 2011).*

Line 14, add "the" between by and different

Done

Line 19, see comment above, replace "inverse" with "iterative"

Done.

Line 29, the model does not "consider" or is "keeping track" of temperature, the algorithm or post processing does, suggest to rewrite. Suggest to add "computed" after "ice is"

The text has been modified to clarify this:

*SICOPOLIS is able to model polythermal ice sheets, i.e., explicitly identifies potential temperate regions in which the modeled ice temperature is at the pressure-melting point (Greve, 1997).*

Page 5 Line 1, see comment above, "detect" seems a strange selection of word, suggest to replace with "determine" or "identify"

Done.

Line 14, not clear wording "a consistent use of inverted distributions of Co" - (replace inverted with determined, or calibrated)

Sentence deleted (this info is a duplicate of the beginning of paragraph).

Line 21, not clear wording: "enters the computation of SStA velocities" – suggest some- thing like "and SStA velocities are computed for this point"

Done. The text has been modified as suggested:

*If r is larger than the threshold, that grid point is flagged as streaming ice and SStA velocities are computed for that point.*

Line 25, suggest to delete "which is assigned to the SStA" this is explained in next sentence (and add velocity after SstA)

Done.

Page 6, Line 2, suggest to add "SIA and SstA" before "Velocities are . . ."

Done.

Line 16-17, unclear wording, suggest something like "It is rather used to determine. . .. . . the computed SStA contribution should partly or completely replace sliding"

Reworded as suggested:

*It is rather used to determine how much the computed SStA contribution should replace the basal velocities used to compute the SIA solution.*

Line 24, sentence is not clear, suggest to rewrite, something like "In the continental interior the modelled ice flow is dominated by the SIA solution

Modified as suggested:

*This approach is based on the assumption that on ice shelves the SIA contribution is negligible due to low surface gradients, and therefore the modelled ice flow is dominated by the SStA solution, whereas in the continental interior the modeled ice flow is dominated by the SIA solution.*

Line 29 see comment above, replace "inversion" with "determination or calibration"

Done.

Line 30, replace "infer" with "determine"

Done.

Page 7, Line 11, see comment above, "easier activation of the inversion procedure" suggest to rewrite to something like "more frequent computation of sliding velocity" - what does "slightly" mean here?

The text has been modified to clarify this:

*In contrast to previous studies using SICOPOLIS where $\gamma=1K$, this parameter is set to 3K, which allows for a more frequent calibration of the sliding coefficients (Pollard and DeConto, 2012).*

Line 14, suggest to delete "similarly, and"

Done.

Line 16, suggest to replace "speed" with "velocity", not clear what "local adjustment" means here, elevation, or Co? suggest to replace "keeps the inversion" with "prevents the method"

Done. The text has been modified to clarify this:

*Additionally, we implemented the following condition: When the computed surface ice velocity reaches an ancillary speed limit at a certain grid point, the adjustment of $C\_0$ for that point is halted.*

Line 17 suggest to replace "over-adjustment of" with "over-adjusting", replace "speed" with "velocity"

Done.

Line 18, suggest to delete "the" before "numerical"

Done.

Line 20, replace "This" with "These" (plural of values)

Done.

Section in lines 22-29 is not clear, and needs rewriting for clarification what is the time in the iterative method or calibration?

Done. We have modified the paragraph to clarify this. The new version reads:

*The iterative technique involves an additional limiting condition that prevents over-adjustments of C_0. At each time step and individually for each grid point, if the adjustment implemented at the previous time step reduces the difference between the modelled and observed ice thickness, the adjustment is skipped for the current time step. This allows previous adjustments to fully develop their effects over the following time steps and prevents the technique from adding unnecessary extra adjustments that can result in overshoots. The calibration is reactivated when the time derivative of the modeled ice thickness becomes zero (i.e. the difference between modelled and observed is not reduced anymore) or the misfit starts increasing (e.g. due to increased influx from surrounding areas). Our experiments have shown that this additional feature enables the use of a smaller Δt_inv (50 years here compared to 500--10,000 years in Pollard and DeConto (2012)), because further adjustments will only be applied when and where strictly necessary. A further benefit is that it indirectly allows non-local adjustments of C_0 influence the local ice dynamics: If an adjustment applied in the vicinity of a grid point reduces its misfit, further adjustments at that grid point will still be halted.*

Line 24, "time derivative of the ice thickness", do you mean observed or modelled?

We mean modeled. We have modified the paragraph to clarify this (see previous point)

why suspend adjustment if previous time step reduced the difference?

Because the previous adjustment can still affect the evolution of the ice sheet during the next time steps. In other words, it could contain the exact amount of adjustment needed for the best fit, requiring just a few extra steps to minimize the difference. Without this, the algorithm would only check for the magnitude of

the difference, adding a potentially unnecessary extra adjustment that could result in an overshoot. If the previous adjustment is not enough, the small time step used (50 years, which is not possible without our algorithm) will ensure a prompt correction. We have modified the paragraph to clarify this (see previous points).

(line 24): What process?

We mean the calibration. We have modified the paragraph to clarify this:

*The calibration is reactivated when [...]*

Line 26 replace "overshoot prevention" with "over-adjustment of Co"

Done.

line 27, suggest to replace "lets" with "allows" and delete "to" before "influence"

Done.

line 31, add "the" before "fringing"

Done.

Page 8 See comment above, move lines 6-18 to a Data description section

Done.

Line 6, re- place "which" with "that"

Done.

Line 19 suggest to add "is used to" before "account", replace "changes" with "discrepancies" , replace "by" with "with"

Done.

Line 23, see comment above, replace "inversion" with "iterative" or calibration

Done.

Line 24, this is forward method, suggest to replace "inversion" with "iterative"

Done.

Line 28, replace "inversion" with "calibration" see above

Done.

Line 30, suggest to delete "one-to-one"

Done.

Line 33, suggest to replace "not accounted for" with "is not included in the simulations"

Done.

Page 9 Line 1, see above, replace "inversion" with "calibration"

Done.

Line 12, replace "simulations" with "calibration run"

Done.

Line 13, see above, replace "inverse technique" with "iterative" or calibration

Done.

Line 27 what do you mean by "glacier flux gate" do you mean  an outlet glacier?

We mean the area where an ice stream reaches the grounding line, usually exhibiting very fast ice flow. We have changed the text to clarify this:

*These mainly occur close to the ice sheet margins where an ice stream reaching the grounding line is often represented by only one grid cell at low resolution*

Line 31 not clear text "which overlap at their interface" needs clarification here

We have changed the text to clarify this:

*11 equidistant grid points for temperate ice and 81 grid points for ``cold'' ice densifying towards the base, sharing the grid point at their interface.*

Page 10 Line 8, suggest to add "a" before "new"

Done.

Line 12-13, this sentence is not clear and needs rewriting (what is internal operation of the hybrid schemes?)

We have changed the text to clarify this:

*In addition, the influence of variations in the parameters controlling how each scheme combines the SIA and SStA velocities is assessed for a wide range of parameter values.*

Line 17, what does "quasi-equilibrium" mean here?

Replaced by "equilibrium", used and defined below (see next point)

Line 19, what is "negligible" in this context? A percentage or some value would be useful here

We consider negligible a change over a prolonged time (>10000 years) smaller than 0.01 %. We have added this to the text to make it clear.

Line 24, suggest to replace "prevents" with "does not require"

Done.

Page 11 Line 9, the smallest error of 49.9m is according to the table using HS-1, is there an error in the table? If not then this section should be rewritten to reflect that.

We have updated the table to reflect the increase in grid resolution and modified the paragraph accordingly.

Line 21, replace "independently" with "independent

Done.

- here some discussion would be appropriate about if this is related to the common SMB forcing or geothermal heat flux?

Done. We also included the potential influence of the uncertainties in the topographic data, as described above (see related specific comment).

Line 22, add "simulated" before "frozen"

Done.

Line 24, suggest to edit, change to something like: "far below the pressure melting point" and delete "the" before "white coloured"

Done as suggested.

Line 25, edit the text, delete "on the other hand", suggest to write "Areas where ice is underestimated . . ." – but what does "sparsely distributed" mean?

Done. We have changed the text to clarify this:

*Areas where ice thickness is underestimated are mainly located at and around the ice margins [...]*

Line 27 replace "inversion" with "determination" or "computation"

Done.

Line 31, replace "inversion" with "iterative" or calibration

Done.

Page 12 Line 2, replace "inverted" with "applied" or "computed"

Done.

Line 4-5, same comment, suggest to write "region where Co is not applied"

Done.

Line 6, replace "inversion" with "calibration"

Done.

Line 21, delete "also"

Done.

Line 26, what do you mean here? What significant modification of Pollard and DeConto (2012) scheme would result in similar values of Co?

We mean that if Pollard and DeConto (2012) used a different hybrid scheme, we would expect a similar degree of variation in their calibrated values of $C_0$. This line duplicates what is implied in the first sentence of the paragraph, and therefore it has been deleted to avoid confusion.

suggest to replace "perturbation" with "distribution" or "pattern" and delete "inverted"

Done (see previous point).

Line 30 add "the" before hybrid

Done.

Line 33 replace "small" with "low" and add "the" after "near"

Done.

Line 35, what are "high velocity flanks"? suggest to add "sheet" before "margins" and at the end of line, after "ice"

We mean the fast flowing ice streams reaching the grounding line. We have modified the text to clarify this and added the corrections:

*distinguishing between ice sheet areas with low velocities near the ice divides and fast flowing ice streams reaching the ice sheet margins*

Page 13 Line 2, strange wording, suggest to replace "contaminated by" with "characterized with"

Done.

Line 6, add "the" before hybrid

Done.

Line 7, suggest to replace "in" with "at"

Done.

Line 8, "flux gates" - not clear, is this a specific location?

We mean the portion of the glacier closest to the grounding line. We have modified the text to clarify this:

*Furthermore, modelled surface velocities are generally overestimated at the grounding zone of most outlet glaciers*

Line 11, delete "inferred" and delete "On the other hand"

Done.

Line 14, delete "inferred"

Done.

Line 19, delete "a" before slow ice motion, replace "flow speed" with "velocity" and "predicted" by "simulated"

Done.

Line 21, delete "Arguably"

Done.

Line 22, replace "stagnated" with "stagnant"

Done.

Line 25, not clear wording "deviated to either side" - "pushed to merge" are you referring to modelled or observed velocity? – what side of what?, what is pushing what?, suggest to replace "deficiencies" with "errors"

Done. We have modified the text to clarify this:

*In other cases, the modelled rapid ice flow follows a different route compared to observations, sometimes merging with adjacent ice streams. These shifts may originate from local errors in the bedrock topography data accentuated by its projection onto the coarse horizontal grid we use here.*

Line 28, according to the table, it is HS-1 that has the minimum misfit for the ice thickness

Done. The text has been modified to fix this.

Line 30, see above, replace "inversion" with "calibration"

Done.

Line 31, suggest to delete "in the modelled surface velocities"

Done.

Line 33, what do you mean by "opposite ends"?

We mean that their degrees of fit to observations is very different (keeping in mind the limitations of the method, as explained in the manuscript), with SoS performing the poorest and HS-2b representing one of the best fits in terms of final ice sheet geometry. We have modified the text to clarify this:

*Although the results of the HS-2b simulation presented in Section 5 are in many aspects similar to those from the SoS, their respective skills in reproducing observations are very different [...]*

Line 34, suggest to replace "enabling" with "adding"

Done.

 Page 14 Line 4, suggest to replace "imply" with "control"

Done.

Line 5, what does "internal operation of the hybrid scheme" mean?

We have modified the text to clarify this:

*In order to provide a deeper insight into how each hybrid scheme combines the SIA and SStA velocities, [...]*

Line 11, suggest to replace "the velocity field from the observational data set" with " the observed velocity" Line 12, suggest to re- place "simple differentiation" with "transition"

Done.

Line 17, reword, "tends to prevent" to "can prevent" and delete "s" in causes (and cause underestimation..)

Done.

Line 24, what is a "cursory comparison" ? clarification is needed

We mean that is readily visible from the figures. We have modified the text to avoid confusion:

*At a first glance it may seem that overestimations are caused by an excessive contribution of the SStA, but a comparison with the SoS scatter plot shows that this is not necessarily the case.*

Line 29-30 suggest to rewrite (delete Here we attempt) and write: To isolate the influence . . .. we plot averaged errors . . .

Done.

Line 35, replace "inferred" with "resulting" or delete "inferred"

Done.

Page 15 Line 1 add "the" before different

Done.

Lines 2-5 this sentence is not clear and does not explain the difference between the 3 figures in Fig.5 clarification is needed here

Figure 5 has been removed (see previous points).

Line 9, suggest to replace "looking" with "comparing"

Done.

Line 19 delete "the" before "period"

Done.

Line 26 – this line is not clear, what do you mean by "generalizing" – see comment above, it is not clear what information about the model can be drawn from Figures 5 and 6

We have modified the text and removed Figures 5 and 6 (see previous points).

Line 34, delete "On the other hand,"

Done.

Page 16 Line 2 suggest to add "distributions of Co from" before the HS-1 and HS-3

Done.

Lines 1-6, it is not clear from this text what this analysis gives for useful information about the different schemes used in the study

The text has been modified to reflect our answer to the related general comment.

Line 8, add "The" at beginning of line

Done.

Line 11, rewrite: "allows us" is a strange wording here

Done.

*In order to explore the sensitivity of the results to parameter variations within this parameter space, we perform an additional series of experiments where we vary the somewhat arbitrary threshold and reference quantities used by some of the hybrid schemes.*

Line 23, what is the criteria  for selecting lower and upper limits for each scheme?

For the HS-1 values outside the [0,1] interval are invalid. For HS-2a and HS-2b, our tests showed that higher or lower values produce no noticeable differences compared to the representative limits we chose. We have modified the text to clarify this:

*Here we test parameter values within a range that contain almost every possible scenario, either because values outside the range are unphysical (HS-1) or they*

*exhibit no noticeable differences compared to the range limits (HS-2a and HS-2b).*

Line 31, suggest to replace "decrease" with "get worse"

Done.

Page 17 Line 4, delete "by" before 2.5%
Done.

Line 24, delete "," after "enables"

Done.

Line 27-29 suggest to rewrite, something like "differ in the way ways the 1) relative contributions . . . are computed 2) areas where . . . is applied is determined, and 3) basal sliding is accounted for.

Done.

Line 30, see above, replace "inverse" with "iterative" or "calibration"

Done.

Line 31, delete "all" or add "the applied" after "all"

Done.

Line 32, add "the" before "schemes", replace "below" with "less than" and add "the" before "total"

Done.

Page 18 Line 3, delete one "can"

Done.

Line 5, add "the hybrid" after "all"

Done.

Line 7, add "the" before hybrid

Done.

Line 8, suggest to replace "develops" with "exists"

Done.

Line 15 What particular scheme is discussed here? It is not clear

Done. We have modified the text to clarify this:

*We have found that the HS-2a tends to increase basal sliding coefficients in an attempt to compensate for an insufficient sliding that would otherwise lead to a larger misfit.*

Line 19 add "," after Here

Done.

Line 27, what does "the neglect of paleoclimate signal" mean here – can applied geothermal heat flux, or applied SMB play a role here?

We mean the neglect of transient temperature effects from previous glaciations and the glacial isostatic adjustments in the bedrock. In the areas mentioned in the text, the geothermal heat flux mostly allows the schemes to calibrate the sliding coefficients, providing the good fit mentioned in the text. These processes were tested by Pollard and DeConto (2012), finding only a small on the results. A dryer climate forcing over these areas would in fact reduce the velocities, because less mass would need to be removed in order to obtain the same fit. We have modified the text to include these points:

*Such misfits might originate from other factors such as, e.g., the lower model resolution, the assumption of an ice sheet in equilibrium, or biases in the model-based geothermal heat flux and/or climatic forcing data sets (Pollard and DeConto, 2012).*

Line 33, does the scheme affect the way the temperature is computed?

Not explicitly. However, due to the coupling of the evolution equations the computed velocities do affect the ice temperature. From the four hybrid schemes, only the HS-1 shows a noticeable difference in the basal temperature field, due to its particular threshold-based selection of grid points where the combination of SIA and SStA is applied. We have added a discussion of this point in the conclusions, which contributes to our evaluation of the applicability of each hybrid scheme in future work (see first point)

Line 34, does the scheme allow determination of hard or soft bed? – it can only indicate high or low value for Co, or what?

We agree. The calibration of the sliding parameter is just an attempt to quantify the combined effect of several processes influencing whether and how the ice slides over the bed. We have modified the text to clarify this:

*Furthermore, there is a qualitative agreement in the patterns of low vs. high values of C_0 obtained from each calibration run.*

Page 19 Line 3, suggest to replace "inferred" with "calibrated"

Done.

- not clear how high variability can "provide an opportunity to quantify the effects of the uncertainty" – suggest to rewrite to clarify what is meant here.

We have modified the text to clarify this:

*We assessed the effects of the high variability in the calibrated parameter distributions derived from different hybrid schemes by performing additional experiments in which averaged and swapped distributions of basal sliding coefficients are prescribed as external data sets.*

Line 8, see comments above, replace "inverse method" with "iterative technique"

Done.

Line 9, what is meant with "internal consistency required to avoid misfit" suggest to rewrite to clarify.

Done (see answer to related general comment)

Suggest that the concluding sentence of the paper will state the main results of the study and how it can be useful for further modelling approaches.

Done (see answer to related general comment)

Figures and tables

Table 2, suggest that the caption include some explanation, referring to text, what the different lines stand for (HS-1 etc).

Done.

See comments above, the lines may have got mixed up, since the text states that the minimum difference for elevation is for HS-2, but 49.9 m is in line HS-1

Done, we have checked this (see answer to comments above)

Figure 2, is this at the end of the simulation?

Yes. We have modified the caption to clarify this:

*Comparison of the equilibrium ice sheet states derived from different schemes at the end of the simulations. [...]*

Figure 4, suggest to replace "grid cell" with "point" for the right hand column figures

Done.

Figure 5, the figure caption is not clear and needs editing. What does "quantify different ice flow regimes" mean here? The y-axis label is mean velocity ratio, but the text states Surface velocity error, clarify what is shown here. Suggest to replace "by" with "as" before ratios.

This figure as been removed (see previous points)

Figure 6, as discussed above it is not clear how this figure is useful. What information can be gained from this analysis? "retrieved distributions of basal sliding coefficients" is not clear, do you mean standard deviation of the determined values with each scheme in each grid point?

This figure as been removed (see previous points)

Figure 7, what does this figure tell us? What meaning does the median of the inferred distributions of Co have? Is this a useful quantity?

Please see answer to related general comment

Figure 8, what do the bumps in the lines of HS-1 with r-thr=0.0 and HS-2b for v-ref=1000 m/yr shortly after 50 kyr mean? Is this instability in the simulations?

Yes, it corresponds to instabilities in the simulations. We discuss this in the conclusions (see also answer to first general comment)

Figure 9 "throughout the simulations" – do you mean at the end of simulations?

Yes. We have modified the caption to clarify this:

*Mean differences between the modelled and observed ice thickness at the end of the simulations, [...]*

**Cited studies:**

Fretwell, P., Pritchard, H. D., Vaughan, D. G., Bamber, J. L., Barrand, N. E., Bell, R., ... & Catania, G. (2013). Bedmap2: improved ice bed, surface and thickness datasets for Antarctica. The Cryosphere, 7(1).

Pollard, D., & DeConto, R. M. (2012). A simple inverse method for the distribution of basal sliding coefficients under ice sheets, applied to Antarctica. *The Cryosphere*, *6*(5), 953-971.

---

## Author Comment (AC2) · 4 Oct 2016

Responses to review 2 of the manuscript "Comparison of hybrid schemes for the combination of Shallow Approximations in numerical simulations of the Antarctic Ice Sheet ":

The manuscript by Bernales et al uses an ice-sheet model for Antarctica to test different schemes for the calculating of the ice velocities. The main focus is on combining the shallow ice and shelfy stream approximations and use different hybrid schemes to combine the two. Four different schemes are implemented in the model and used to inversely determine the basal sliding coefficient C0 by looking at the difference between modelled and observed ice thickness. Differences between the four schemes are explained clearly, through how the spatially varying basal sliding coefficient differ for each scheme, how grounded ice volume evolves and how ice thicknesses vary relative to the observations. Model performance is also checked against observations of ice surface velocities. Methods and results are explained clearly. The two main concerns I have are that the manuscript does miss a real strong final conclusion and that uncertainties in the methodology should be discussed in more detail.

We thank the reviewer for their very constructive comments that helped to strengthen the manuscript. Our replies can be found below.

**Main comments**

1. Uncertainties in methodology

Paragraph starting on Page 2, Line 30 to Page 3, Line 8:

You should here also mention that there are more parameters and/or processes involved that are poorly constrained (or less known) such as bedrock uplift due to post- glacial rebound (GIA), response of the bed itself to current changes in ice, heat flux from the bed, properties of ice flow (flow parameter) or the bedrock height uncertainty (see Bedmap2 paper). Following on the uncertainties, an important point to make using this method is that by solving for the basal stress you actually compensate/correct for other (some of them mentioned above) uncertainties in the boundary conditions and uncertainties/errors in the model itself. This is I think a key point to keep in mind when using the methods described in the manuscript. This should be mentioned here and discussed thoroughly in the final discussion/conclusion as well!

Following the suggestion of the reviewer, we have added a discussion of potential effects of these poorly constrained processes and boundary conditions on the modeling results. We have also added a separate Data section where the uncertainty in the bedrock topography data is spelled out. The possibility of canceling errors is a clear limitation of the method, since at present there is no easy way to differentiate what could be an artificial compensation of, e.g., an insufficient model representation of physical processes or deficiencies in the input data sets from the actual conditions at the base of the ice sheet. This shortcoming will hopefully be overcome in the future once the necessary observational data sets and more sophisticated ice models become available. For now -and for the purposes of this study-, the risk of error cancellation is over-passed by the dramatic reduction in the misfit between the modeled and observed ice thickness and surface velocities. This limitation is now discussed in the method description and in the final discussion sections.

2. Rewording of basal velocities and SIA

Starting on Page 4, Line 16:

Throughout the manuscript it is mentioned that SIA has a basal velocity. This is mostly a matter of rewording some of the text, not so much an error but rather how this is actually computed. I think the SIA itself does not have a basal velocity, but rather that the basal velocity is a basal boundary condition to calculating the SIA, which could also be set to zero if you do not include sliding at all. This is actually nicely illustrated in Fig. 1 of Bueler and Brown (JGR, 2009). I think this could be rewritten in some parts of the manuscript. In the general comments below some of these lines are mentioned.

We agree. All instances have been rewritten accordingly.

3. Final conclusions

On page 19:

The sentence "This suggests .. Stokes equations." seems quite obvious but good to mention. However in the final conclusions I do miss a discussion on two things particularly: - Is a distribution of basal parameter C0 derived using present-day forcing and observations also applicable for long-term paleoclimate simulations, can it be used for other time periods and climate regimes?

This is an interesting and challenging question, since the current availability and the accuracy of the "observational" data for past periods is not sufficient for testing the performance of the reconstructed coefficients under paleoclimate scenarios. Basal thermal regimes of continental-scale ice sheets and thus the areas where model calibration procedures can be applied do not only depend on the external thermal forcing that remains nearly constant in time (e.g., geothermal heat flux) but also on climate conditions and ice sheet geometrical settings that had undergone significant changes in the past (Bentley et al., 2014). There are regions where basal sliding is not identified by the method under the present-day conditions, where paleo-ice streams could have developed under a different climate regime, and vice versa. Thus, the derived distribution of C_0 is not necessarily applicable everywhere. However, we believe that it is a good first-order approximation, and is definitely a better guess than commonly used single, homogeneous values of C_0 over the entire domain. The method used to derive C_0 could be potentially extended to the use for other time periods/climate regimes (or even adapted for the transitions between largely dissimilar regimes), providing that the necessary topographic and climate data are available. Looking at the recent efforts to reconstruct the past geometries of the Antarctic Ice Sheet (Mackintosh et al., 2014, Bentley et al., 2014), this may become well possible in the upcoming decades. We have added this point in the final discussion.

- What is the best scheme to be used here? Is there a scheme that simulates the observations best, also considering the lowest over- and underestimations, i.e. the lowest absolute difference in ice thickness.

We have evaluated the performance of each scheme based on both its fit to observations and numerical stability for a range of model setups. We have found that at higher grid resolutions the HS-1 and HS-2b schemes become numerically unstable, due to large gradients in basal sliding coefficients arising from the use of basal velocities as boundary conditions for the SIA solutions in conjunction with the calibration of sliding coefficients. The HS-2a and HS-3, which utilize the SStA as a sliding law, are numerically more stable to variations in model parameters and changes in grid resolution (especially HS-2a, which rarely produces simulation crashes). The drawback of the HS-2a is in its limited ability to

influence the fit between the modeled and observed ice thickness in ice sheet sectors where the SStA velocities are low (<100 m/yr), which is the case over large tracts of the Antarctic interior. The HS-3 overcomes this limitation by accounting for the SStA contribution everywhere, but so it does with the SIA velocities, which in certain areas such as, e.g., the steep ice sheet margins are excessively high, as shown in Figure 3 of the original version of the manuscript for the SoS. To improve the performance of both schemes, our future work will reconcile the drawbacks of HS-2a with the advantages of HS-3, providing a very stable and flexible hybrid scheme. We have added this point in the final discussion.

**General comments**

Line 1: Replace introduced with used or implemented

Done.

L 5: What do you mean with realistic scenario? Do you mean like present day climate forcing? Please be specific.

We mean a non-synthetic/simplified ice sheet geometry. We have modified the text to avoid confusion:

*Here, we implement four different hybrid schemes into a model of the Antarctic Ice Sheet in order to compare their performance under present-day conditions.*

L 8: Remove comma after Despite this

Done.

L 8: Robust agreement with what, or of which variable? Again please be specific.

We have modified the text to clarify this:

*[…] we observe a robust agreement in the reconstructed patterns of basal sliding parameters.*

L 17: Change to: However, the time scales over which an ice sheet builds up and disintegrates

Done.

L 9: Remove commas before membrane and after important

Done.

L 7-14: You could first introduce the SSA (L10-14) and then discuss the ice streams (L7-10).

Done as suggested.

L 15: Instead of mentioning highly dynamic regions, actually state here something like: The limitations

of SIA models for calculating the highly dynamics ice streams have ..

We have modified the text as follows:

*The limitations of SIA models for reproducing the dynamics of ice streams have prompted the development of [...]*

L30: Rephrase realistic scenarios, same as in the abstract, present-day climate forcing?

We have modified the text as suggested:

*Despite the above differences among models, all of them are subject to common limitations when applied to the present-day Antarctic Ice Sheet*

L 17: No new paragraph, start sentence as: First, the ice-sheet model

Done.

L 16: Please rewrite, the SIA itself does not have a basal velocity, but rather U-b is here the basal boundary condition to calculating the SIA.

We have modified the text as suggested:

*At the base of the grounded ice sectors, bedrock stress conditions and the associated potential for sliding are linked to the basal velocity, u_b, used as a boundary condition for the computation of the SIA velocities, through an empirical Weertman-type sliding*

L 18: Eexplain here how the effective basal pressure Nb is calculated.

Done. The description reads:

*$N\_b$ is the effective basal pressure, computed as $N\_b = rho_{ice} \, g \, H - rho_{sw} \, g \, H_{sw}$, where $rho_{ice}$ and $rho_{sw}$ are the density of ice and sea water, respectively, g is the gravitational acceleration, H is the modelled ice thickness, and $H_{sw}$ is the difference between the mean sea level and the ice base topography.*

L 22: Is Tm an actual temperature or a temperature difference. Explain this in the text.

We have modified the text in page 4, line 5 to clarify this:

*[…] $T\_m$ is the temperature difference relative to the pressure-melting point, [...]*

L 19: Rewrite to something like: where ub is the Weertman sliding velocity (Eq. (2)) and us the surface SIA velocity, respectively.

We have modified the text as suggested:

*where u_b is the Weertman sliding velocity (Eq. 2) and u_s is the surface SIA velocity.*

L 11-16: Do not mention sliding-SIA but rather SIA including (Weertman) sliding. Here it could also be discussed the carefulness of using Weertman sliding, as discussed by Bueler and Brow (JGR, 2009; see also their Appendix B) it leads to a discontinuities in the velocity field.

We have modified the text as suggested:

*As described in Section 2.1, the SIA solution in SICOPOLIS is computed using the Weertman sliding component coming from Eq. 2 as a boundary condition. To assess the influence of a SIA solution including Weertman sliding, we have divided this hybrid scheme into two: A sub-scheme (HS-2a) that replicates the idea of Bueler and Brown (2009) and prescribes no basal velocities in the computation of the SIA velocities, and a sub-scheme (HS-2b) that keeps the Weertman sliding component and uses it to compute a slightly modified weight [...]*

We have mentioned the discussion on Weertman-type sliding in the description of the model with the respective reference to Bueler and Brown (2009).

L 16: change to: where ub is the basal sliding velocity as in ..

Done.

L 28: At the end of section 2.2 a table could be added that summarises the 4 schemes with the column show something like : 1) name, 2) sliding 3) reference to equations, 4) reference to studies

Done. We have added the table as suggested.

L 23-24: Can you explain here and perhaps clarify in the text why you stop the adjustment when the difference between modelled and observed ice thickness is reduced?? A reduction of this difference is actually what you want right?

Because the previous adjustment can still affect the evolution of the ice sheet during the next time steps. In other words, it could contain the exact amount of adjustment needed for the best fit, requiring just a few extra steps to minimize the difference. Without this, the algorithm would only check for the magnitude of the difference, adding a potentially unnecessary extra adjustment that could result in an overshoot. If the previous adjustment is not enough, the small time step used (50 years, which is not possible without our algorithm) will ensure a prompt correction. We have modified the paragraph to clarify this (see previous points).

The text has been modified to clarify this:

*The iterative technique involves an additional limiting condition that prevents over-adjustments of C_0. At each time step and individually for each grid point, if the adjustment implemented at the previous*

*time step reduces the difference between the modelled and observed ice thickness, the adjustment is skipped for the current time step. This allows previous adjustments to fully develop their effects over the following time steps and prevents the technique from adding unnecessary extra adjustments that can result in overshoots. The calibration is reactivated when the time derivative of the modeled ice thickness becomes zero (i.e. the difference between modelled and observed is not reduced anymore) or the misfit starts increasing (e.g. due to increased influx from surrounding areas). Our experiments have shown that this additional feature enables the use of a smaller Δt_inv (50 years here compared to 500-- 10,000 years in Pollard and DeConto (2012)), because further adjustments will only be applied when and where strictly necessary. A further benefit is that it indirectly allows non-local adjustments of C_0 influence the local ice dynamics: If an adjustment applied in the vicinity of a grid point reduces its misfit, further adjustments at that grid point will still be halted.*

L 22-29: This paragraph could be replaced to Section 3.

We prefer to keep this paragraph here, since it is an important part of the calibration algorithm and follows the description of other constraints in the method.

L 13: On spin up procedures for ice-sheet models see also this paper, you might want to refer to this. Fyke, J. G. and Sacks, W. J. and Lipscomb, W. H., A technique for generating consistent ice sheet initial conditions for coupled ice sheet/climate models. Geoscientific Model Development, 7, 1183 – 1195, 2014.

We have added the reference as suggested.

L 15-16: Reword: a simulation time of 100.000 years (100 kyr)

Done.

L 21-22: Spanning 100 kyr (mention exact length of your experiments)

Done.

L 1: Remove -and get insight into-

Done.

L 4: Be specific: observed ice-sheet thickness and surface velocities as a measure ..

Done.

L 19: .. the variations of the total grounded ..

Done.

L 26: Numerically more or equally stable, and also a similar computational time com- pared to the SoS?

As explained in the paragraph. All hybrid schemes take longer than the SoS due to the calculation of SStA velocities over a larger domain. However, the hybrid schemes require less of these (in this case, very long) iterations to get convergence in the solution, which makes them less prone to numerical instabilities compared to SoS. We have modified the paragraph to clarify this:

*Compared to the SoS, the computation of SStA velocities in grounded ice sectors implies an extra computational effort for each iteration in the numerical solvers, with the computing time increasing by a factor of ~4 for the applied hybrid schemes. The computing time of the HS-1 is somewhat shorter relative to the other hybrid schemes (but still longer than for the SoS) due to the prognostic identification of ice streams that does not require a computation of SStA velocities over the entire ice sheet. However, we have observed that the iterative solvers in the model require a substantially smaller number of iterations when the hybrid schemes are used, making them numerically more stable compared to the SoS.*

L 1: Change to: that include basal sliding with the SIA solution

Done.

L 4: Remove -high frequency-

Done.

L 7: The inset of Figure 1

Done.

L 13: Change to: that include basal sliding with the SIA.

Done.

L 20-21: The large overestimation of ice thickness between Shakleton Range and the Pensacola Mountains as you state is seen in all panels of Fig. 2. Could you give a more clear explanation why this particular area is overestimated. Also note that this is a region of large uncertainties in ice thickness (see Bedmap2 paper).

As you have pointed out, this area has one of the largest uncertainties in the BEDMAP2 data. The lack of observational data produced an unrealistically smooth bedrock topography (Fretwell et al., 2013). We attribute to this uncertainty the absence of  topographically driven ice streams that fosters the accumulation of ice at the interior. We have added this point to the text.

L 28, 32: I suggest not to refer to Fig. 1a but to: the inset of Fig. 1.

Done.

L 2: Change to: over the areas where there is no sliding and C0 is not inverted.

Done.

L 32: Change to: .. that are unresolved by the model due to its..

Done.

L 34: Remove comma after divides.

Done.

L 7: Remove commas before and after for example

Done.

L 26: Change to: .. projection onto the course horizontal grid we use here.

Done.
L 29: Remove comma after simulation

Done.

L 33: Change to: Equation (10) uses the basal velocities from equation (2) to compute..

Done.

L 34-35: Where sliding velocities from equation (2) are high.

Done.

L 1: Here also perhaps mention how H-2a compares to H-2b?

Done.

L 5: are implemented

Done.

L 5: Change to: .. hybrid schemes, Fig. 4 illustrates the

Done.

L 22: Remove comma after general

Done.

L 23-24: At a first glance it may

Done.

L 25: Change to full stop: the case.

Done.

L 2: Change to: The misfit for Hs-1 increases

Done.

L 4: The Hs-3 experiment shows an ..

Done.

L 19: Change to: the basal velocity in the Hs-2b experiments

Done.

L 24: Remove commas before and after continental-scale.

Done.

L 25: Remove entire

Done.

L 1: Also mention the range of the ice thickness errors not only the means.

Done.

L 17: Change to: which adds up SIA (without sliding) and SstA.

Done.

L 29: Also mention the uncertainties in ice thickness and surface velocities (see their respective references).

Done.

Tables and Figures

Table 2:
Should the sum of the last two columns not be 100%? Please clarify and explain.

The columns show the percentage of area where the flow is predominantly dominated either by the SIA or the SStA, defined as the area where the weight w is <0.25 (SIA-dominated) or >0.75 (SstA dominated). Areas where w is between 0.25 and 0.75 are excluded from these percentages, since they present a closer mix of both velocity solutions. We have clarified this in the caption.

Fig. 1:
- Label a and b can be removed, rather refer to fig. 1 and inset of fig. 1.

Done.

- Instead of a coloured bar for the pre-initialization the names of the periods could also be placed above (pre-initialization period and automatic calibration).

Done.

- In the legend of the inset the difference between solid and dashed should be clearer.

Done.

Fig. 4:
It is not really clear which scheme represents the surface velocities the best. Could you also add a correlation/R-square value to the scatter plots? Also use the correlation in the discussion.

Thanks for the suggestion. We have added the root-mean-square error to the scatter plots, which provides a simple way to clarify this. This information is also used in the results and final discussion sections to aid the determination of the "best" scheme overall.

Fig 8: Hs-2a always seems to be have a misfit, whereas the other scheme should have a possible solution which closely fits the observations, for a particular (probably not the same) set of parameters. Please explain and discuss this in the conclusions.

HS-2a prescribes zero basal velocity before the computation of the SIA. Sliding comes from the SStA, but only in those regions where the SStA velocities are high (see the weights w in Figure 4). Thus, at the continental interior the contribution from the SStA is small and not enough to prevent the overestimations of ice thickness that produce the misfit observed in Figure 8. The HS-3 does not scale the SStA contribution with any weight, and thus is able to prevent the overestimations. This has been added to the final discussion (see also our reply to the third main comment).

**Cited studies:**

Bentley, M. J., Cofaigh, C. Ó., Anderson, J. B., Conway, H., Davies, B., Graham, A. G., ... &

Mackintosh, A. (2014). A community-based geological reconstruction of Antarctic Ice Sheet deglaciation since the Last Glacial Maximum. Quaternary Science Reviews, 100, 1-9.

Bueler, E., & Brown, J. (2009). Shallow shelf approximation as a "sliding law" in a thermomechanically coupled ice sheet model. Journal of Geophysical Research: Earth Surface, 114(F3).

Fretwell, P., Pritchard, H. D., Vaughan, D. G., Bamber, J. L., Barrand, N. E., Bell, R., ... & Catania, G. (2013). Bedmap2: improved ice bed, surface and thickness datasets for Antarctica. The Cryosphere, 7(1).

Mackintosh, A. N., Verleyen, E., O'Brien, P. E., White, D. A., Jones, R. S., McKay, R., ... & Miura, H. (2014). Retreat history of the East Antarctic Ice Sheet since the last glacial maximum. Quaternary Science Reviews, 100, 10-30.

---

## Referee Report (RR1)

Review of manuscript "Comparison of hybrid schemes for the combination of Shallow Approximations in numerical simulations of the Antarctic Ice Sheet" by Jorge Bernales, Irina Rogozhina, Ralf Greve and Maik Thomas [The Cryosphere Discuss. Doi:10.5194/tc-2016-117]

General comments

This paper describes a study of 4 hybrid schemes, combining the Shallow Ice and Shelfy Ice approximations for Antarctic Ice Sheet simulations comparison with Shallow Ice approximation only is also done.  The 4 schemes are implemented into the open source ice sheet model Sicopolis (Greve, 1997), and a number of simulations are made for the Antarctic Ice Sheet.   The paper is clearly written and the conclusions are clear

Specific comments:

The description of the study of model sensitivity to resolution is not clearly written and the higher resolution is not given in the result section, it would give the study more weight if it was clearly stated what resolutions were tested (in the result section) and what conclusions were drawn from the sensitivity study.

Technical corrections:

Page 1, Abstract
Line 1, suggest to add "equations" after "force balance" or rephrase the sentense
Line 2, suggest to rewrite to avoid personalizing SIA (SUA alone is not able to….)  something like SIA cannot …
Line 8 and line 9 instead of "parameter distribution" something like "calibrated basal sliding coefficient" would clarify the sentences
Line 11, it is not clear to what limitations is being referred to, rewriting suggested.
Line 11-12, it is not clear what "easily adaptable calibration techniques" refers to here, the text in the abstract should stand alone and some clarification is needed here.

Page 2, line 1 replace "has" with "have" after approximations
Line 5, suggest to replace "usually described" with something like "can be" or "often"
Line 16, suggest to replace "valid" with "applicable"
Line 20 suggest to replace "heuristically" with "heuristic"

Page 3, line 4, suggest to replace "approaches" with "schemes"
Lines 5-6 something is missing in sentences, something like "as validation" in the end would make sense?
Line 24, here Sicopolis is personalized, suggest to rewrite to something like "is applicable for"

Page 4 line 9 suggest to replace "bedrock stress conditions" with "stress conditions at bed

Page 8, line 7, suggest to replace "we believe" with something like "assume" or "think"

Line 22, suggest to delete "that" after "crust"

Line 28, suggest to rewrite, replace "is not able to" with "cannot"

Line 30-line 2 on page 9, something is missing in description, it appears that PDD method is used to computed the surface melt and accumulation and T is coming from RACMO, why is then a comparison made with surface mass balance in the RACMO paragraph? Is SMB from RACMO used at all in the study? How does the SMB computed with the accumulation and the PDD method applied in this study with the observations?

Line 30, suggest to replace "data" with "model output"

Page 9, line12, how high resolution is tested? See comment above

Page 11, line 13, it is not clear what initialised state refers to here, after 100 ka, or after the full thermal and dynamic equilibria are attained?

Line 31, suggest to add "average" before "error"

Line 32, delete "an" before 83%

Page 12, line 3, suggest to add "absolute" before "error"

Line 5, not clear what "generalized" means here, so you mean "volume change"?

Line 21, it is not clear what is meant by the sentence "areas where it is applied directly", suggest rewriting

Line 27, suggest to replace "recovered" with "estimated"

Line 29 suggest to replace "retrieved" with "estimated"

Page 13, not clear what "relatively higher coverage " means here, suggest to rewrite

Line 20 suggest to replace "regimes" with "velocities"

Page 15, line 22,suggest to replace "distribution" with "values"

Page 16, line 33 suggest to replace "imply" with "results in"

Page 18, line 33, suggest to add "model" after SIA flow

Page 19, line 21, see comment above, how high resolutions were tested, can this sensitivity study be better described?

Line 26, suggest to replace "influence" with "improve"?

Line 33, suggest to add "sliding parameter" after "calibrated"

Page 29, add s at the end of need, "one needs"   also, it is not clear what results "these results" are refereeing to, suggest to clarify

Table 3, suggest to explain in table caption what SoS stands for, so that the table can be read independently

Table 4, also for independence of table suggest to explain what "prescribed median" means here

Figure 2, suggest to add information what happens at each 100 ka time interval, it makes the figure easier to read.  Suggest to add in inset figure "absolute" before "error" and also in figure caption line 2

Figure 7, it is strange that the thick blue line is not in between the thin and dotted blue lines, like the green and the red, is the reference value not between the upper and lower limit? This is strange.

Figure 8, suggest to add "absolute" before "difference in line 1 of figure caption.